# JASPer controls interphase histone H3S10 phosphorylation by chromosomal kinase JIL-1 in *Drosophila*

Christian Albig [1,2], Chao Wang[3], Geoffrey P. Dann[4,6], Felix Wojcik[4], Tamás Schauer[5], Silke Krause[1], Sylvain Maenner[1,7], Weili Cai[3], Yeran Li[3], Jack Girton[3], Tom W. Muir [4], Jørgen Johansen [3], Kristen M. Johansen [3], Peter B. Becker [1*] & Catherine Regnard [1*]

In flies, the chromosomal kinase JIL-1 is responsible for most interphase histone H3S10 phosphorylation and has been proposed to protect active chromatin from acquiring hetero-chromatic marks, such as dimethylated histone H3K9 (H3K9me2) and HP1. Here, we show that JIL-1's targeting to chromatin depends on a PWWP domain-containing protein JASPer (JIL-1 Anchoring and Stabilizing Protein). JASPer-JIL-1 (JJ)-complex is the major form of kinase in vivo and is targeted to active genes and telomeric transposons via binding of the PWWP domain of JASPer to H3K36me3 nucleosomes, to modulate transcriptional output. JIL-1 and JJ-complex depletion in cycling cells lead to small changes in H3K9me2 distribution at active genes and telomeric transposons. Finally, we identify interactors of the endogenous JJ-complex and propose that JIL-1 not only prevents heterochromatin formation but also coordinates chromatin-based regulation in the transcribed part of the genome.

[1] Molecular Biology Division, Biomedical Center, Faculty of Medicine and Center for Integrated Protein Science Munich (CIPSM), LMU Munich, 82152 Martinsried, Germany. [2] Graduate School for Quantitative Biosciences (QBM), LMU Munich, 81377 Munich, Germany. [3] Roy J. Carver Department of Biochemistry, Biophysics, and Molecular Biology, Iowa State University, Ames, IA 50011, USA. [4] Department of Chemistry, Frick Laboratory, Princeton University, Princeton, NJ 08544, USA. [5] Bioinformatics Unit, Biomedical Center, Faculty of Medicine, LMU Munich, 82152 Martinsried, Germany. [6] Present address: Department of Biochemistry and Biophysics, Perelman School of Medicine, University of Pennsylvania, Philadelphia, PA 19104, USA. [7] Present address: UMR7365 CNRS-UL, IMoPA, University of Lorraine, 54505 Vandoeuvre-lès-Nancy, France. *email: pbecker@bmc.med.lmu.de; cregnard@bmc.med.lmu.de

In mammals, several nuclear kinases contribute to phosphorylation of histone H3 at serine 10 (H3S10ph) in interphase, whereas in *Drosophila melanogaster*, the essential kinase JIL-1 is responsible for most of it[1]. The significance of interphase H3S10ph is often underestimated because most H3S10 phosphorylation in asynchronous cell populations stems from mitotic chromatin, where it is deployed by Aurora B kinase [2,3]. Originally, interphase H3S10ph has been associated, in combination with H3K9ac and H3K14ac, with transcriptional activation of immediate early genes upon MAPK activation[4,5]. In *Drosophila*, interphase H3S10ph is enriched at the body of active genes[6]. In mammal, in the extreme case of mouse embryonic stem cells (mESC), ~30% of the genome is enriched for H3S10ph in interphase[7].

The current model assigns JIL-1 to the protection of euchromatin from heterochromatization[8]. According to the phosphomethyl switch model for mitotic H3S10ph[9], placing H3S10ph prevents H3K9 methylation and subsequent binding of heterochromatin components. JIL-1 phosphorylates various H3 peptides with different methylation states, including H3K9me2/3, with comparable efficiency[6], whereas histone methyltransferases of the Su(var)3-9 family are inhibited by H3S10ph[10,11]. Several observations suggest that JIL-1 is important for the balance between euchromatin and heterochromatin. The *Su(var)3-1* alleles of *JIL-1* gene, which lead to the expression of JIL-1 truncated in its C-terminal domain (CTD), result in reduced heterochromatin spreading at euchromatin-heterochromatin boundaries[12,13]. Conversely, in the *JIL-1^{z2/z2}* null mutant, heterochromatin components spread into euchromatin. The spreading of H3K9me2 and HP1 is highest on the euchromatic part of the X chromosome in both sexes[14], the spreading of the 7-zinc-finger protein Su(var)3-7 affects euchromatin similarly on all chromosomes[15]. In addition, JIL-1 phosphorylates Su(var)3-9[16], the histone methyl transferase responsible for H3K9me2/3, suggesting a possible function for JIL-1 at constitutive heterochromatin.

JIL-1 may also play a role at telomeres, which combine features of heterochromatin and euchromatin in *Drosophila*. JIL-1 localizes to arrays composed of the three non-LTR retrotransposons *HeT-A*, *TART*, and *TAHRE* (HTT) on polytene chromosomes in mutants with elongated telomeres[17]. Transcription of HTT arrays is essential for telomere maintenance in flies, and JIL-1 is a positive regulator of retrotransposon transcription[18,19].

At the low resolution of polytene chromosomes, JIL-1 localizes to active chromatin and is enriched on the male dosage-compensated X chromosome[20]. When the binding of JIL-1 to chromatin was studied at higher resolution using chromatin-immunoprecipitation (ChIP), conflicting results were obtained. Our early ChIP-chip study suggested that JIL-1 is found on all transcribed gene bodies and is enriched on X-chromosomal genes in male S2 cells[6]. ChIP-seq experiments from female Kc cells[21] and salivary glands[8] suggested that JIL-1 associates to the 5' end/promoters of active genes and to enhancers.

In this work, we show that the JIL-1 protein level is tightly controlled by JASPer (JIL-1 Anchoring and Stabilizing Protein), a PWWP domain-containing protein. Both proteins form a stable JASPer-JIL-1 (JJ)-complex, the functional form of the kinase in vivo. The PWWP domain of JASPer tethers the JJ-complex to H3K36me3 nucleosomes in vitro. Consistently, the JJ-complex is targeted to H3K36me3 chromatin at active gene bodies and at telomeric transposons in vivo. Depletion of the JJ-complex in flies induces heterochromatin spreading in salivary gland nuclei as described for the JIL-1 deficiency. Using *D. melanogaster* cell lines, we show that depletion of JIL-1 or the JJ-complex modulates the transcriptional output. In male S2 cells, depletion of

JIL-1 results in a modest enrichment of H3K9me2 in the active chromatin, where the JJ-complex binds. Finally, we identify various known and novel interactors of the endogenous JJ-complex, notably chromatin remodeling complexes and subunits of the Set1/COMPASS complex.

## Results

**JIL-1 forms a complex with the protein JASPer.** Since JIL-1 lacks a known chromatin binding domain, we hypothesized that JIL-1 is recruited to chromatin by an interaction partner. To identify such a protein, we used nuclear extracts of *D. melanogaster* embryos to perform preparative immunoprecipitations (IPs) using antibodies against JIL-1. A protein of ~60 kDa co-purified with JIL-1 using two different JIL-1 antibodies (Supplementary Fig. 1a). Mass spectrometry analysis identified the protein as encoded by the gene *CG7946* on chromosome 3R. We named this protein 'JIL-1 Anchoring and Stabilizing Protein' (JASPer). Consistently, reverse IP's using antibodies against JASPer showed that JIL-1 was efficiently co-immunoprecipitated from embryo extracts and with similar efficiency (Fig. 1a). Coexpressing recombinant FLAG-JIL-1 and untagged JASPer[22] yielded a stable complex (Fig. 1b). Coomassie-blue staining suggested a roughly equal stoichiometry for the recombinant and the endogenous complex (Fig. 1b, Supplementary Fig. 1a) (corresponding to a mass ratio of 2.6:1 at calculated molecular weights of 137 kDa for JIL-1 and 53 kDa for JASPer).

JASPer is a well-conserved protein among *Drosophila* species. It has an N-terminal PWWP domain and a C-terminal LEDGF/IBD domain (Fig. 1c, Supplementary Figs. 2a, b). This PWWP-LEDGF domain architecture is found in 94 eukaryotic proteins, with mostly unknown functions, except for the PSIP1/LEDGF chromatin adapter protein, which has pleiotropic functions in HIV infection and cancer development[23,24]. JIL-1 is also well conserved among distant *Drosophila* species (Supplementary Fig. 3), particularly in the N-terminal AGC kinase domain[25], the C-terminal MAPK-related domain and its CTD. The CTD is rich in proline (11%) and arginine (9%) residues and most probably intrinsically disordered. Sequence comparison revealed a prion-like domain (PrlD)[26] and putative PEST sequences[27], which most probably relate to lower stability of the protein because of their intrinsic disorder[28].

Using a LacO-LacI targeting system in flies, we found that LacI-JIL-1 full-length and LacI-JIL-1-CTD recruited endogenous JASPer to the LacO arrays, but JIL-1-ΔCTD did not (Supplementary Fig. 1b). We further mapped the interaction by co-expression and co-purification of various derivatives. Truncations in the CTD of *JIL-1* were designed according to sequence conservation in *Drosophilae* JIL-1 homologs (Fig. 1c and Supplementary Fig. 3). Expression of the C-terminal deletion mutants of FLAG-JIL-1 (mutants a-g) with untagged, full-length JASPer showed that the minimal JASPer binding domain (JBD) encompasses 44 amino acids (982–1025) between the truncations c and d of the CTD (Fig. 1d). The JBD is rich in proline (22%), glutamic acid (16%), and aromatic residues tyrosine/phenylalanine (16%). Furthermore, it contains a stretch of 7 conserved amino acids, DFxGFDE, matching the consensus motif (FxGF) found in proteins interacting with the LEDGF/IBD domain of PSIP1[29]. Indeed, using various JASPer derivatives (Fig. 1c) co-expressed with full-length FLAG-JIL-1, we found that deletion of the 120 amino acids long LEDGF domain in the C-terminal half of JASPer (ΔLEDGF) was sufficient to abrogate binding to JIL-1 (Fig. 1e). This domain contains a high proportion of charged residues (glutamic acid/aspartic acid: 18% and arginine/lysine residues: 17%).

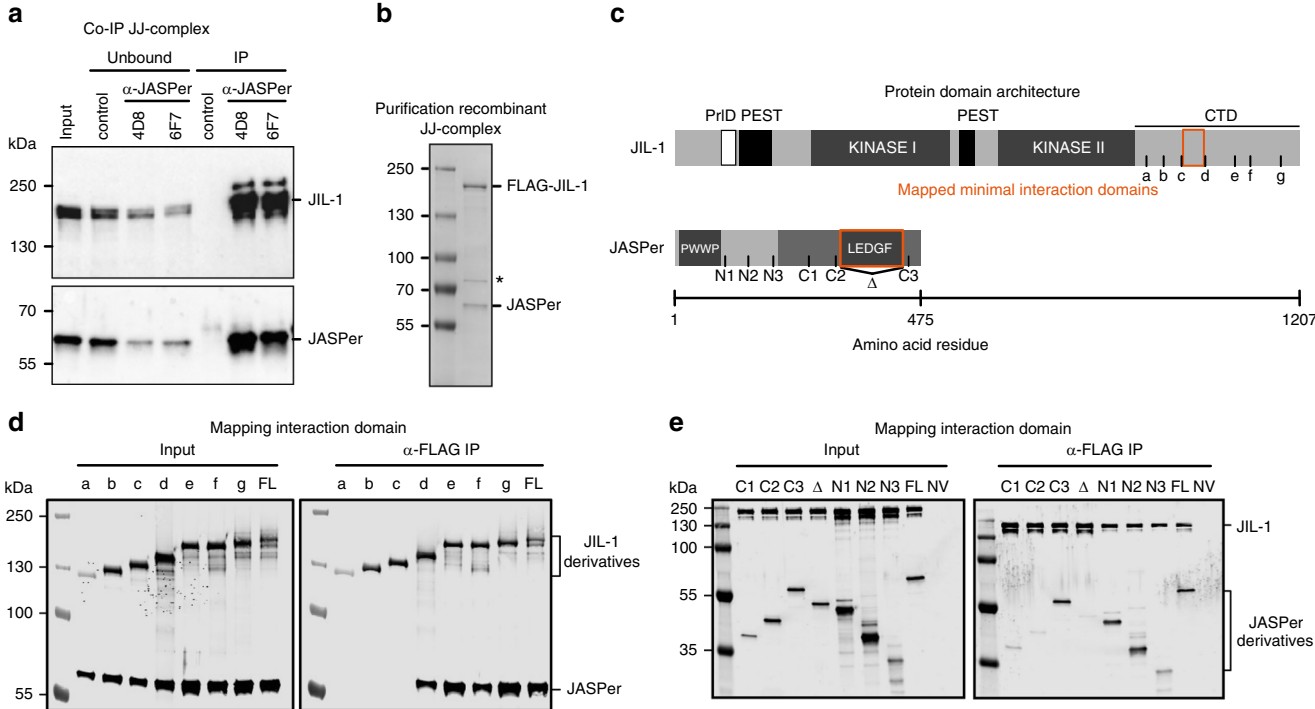

**Fig. 1** JIL-1's C-terminal domain interacts with JASPer's LEDGF domain to form the JJ-complex. **a** Western blot analysis with α-JASPer and α-JIL-1 antibodies of co-IP from nuclear embryo extracts. Co-IP was performed with two different monoclonal α-JASPer antibodies containing culture supernatants and culture medium as control. The corresponding unbound fractions are loaded next to each IP. Molecular weight markers are shown to the left. Source data are provided as a Source Data file. **b** SDS-PAGE with Coomassie staining of recombinant JJ-complex purification from Sf21 cells using a baculovirus dual expression system of FLAG-JIL-1 and untagged JASPer. Molecular weight markers are shown to the left. A contaminant band is marked by asterisk. Source data are provided as a Source Data file. **c** JIL-1 and JASPer domain architecture drawn to scale. In JIL-1, PEST domains are highlighted in black, kinase domains in dark gray and a predicted prion-like domain in white. In JASPer, PWWP and LEDGF domains are highlighted in dark gray and conserved region in intermediate gray. C-terminal truncation breakpoints **a–g** for JIL-1 and N-terminal (N1-N3) and C-terminal truncation breakpoints (C1-C3) for JASPer used in **d** and **e** are indicated. Δ denotes the deletion in JASPer-ΔLEDGF. **d** Western blot analysis using α-JIL-1 and α-JASPer antibodies of co-IP experiments with extracts from Sf21 cells expressing wild type, untagged JASPer and various FLAG-JIL-1 C-terminal deletion mutants. Co-IP was performed with α-FLAG beads. Source data are provided as a Source Data file. **e** Western blot analysis as in d of co-IP experiments with extracts from Sf21 cells expressing various untagged JASPer deletion mutants and FLAG-JIL-1. Uninfected Sf21 cell extract was used as control (NV = no virus). Co-IP was performed with α-FLAG beads. Source data are provided as a Source Data file.

**JASPer stabilizes JIL-1 in vivo**. To understand the function of JASPer in the JJ-complex, we generated *JASPer^cw2* null allele by imprecise excision of the P-element in an appropriate EP line. The deletion encompassed the coding region of both described transcripts (Fig. 2a). Analysis of the salivary glands of homozygous *JASPer^cw2/cw2* mutants showed that JASPer was not detectable by western blot and on polytene chromosome spreads (Fig. 2b, c). Remarkably, JIL-1 was also not detectable in the absence of JASPer. The lack of the JJ-complex in turn correlates with undetected phosphorylation of H3S10, confirming that kinase as the major source of this modification in interphase (Fig. 2d). Such a direct relationship between kinase and H3S10ph cannot be seen in an exponentially growing cell population, due to the strong dominance of mitotic H3S10ph[6]. As also described for the *JIL-1^z2/z2* hypomorph mutant[14], global H3K9me2 levels were unchanged (Fig. 2b) but the mark redistributed from the chromocenter to the euchromatic chromosomal arms, in particular of the X chromosome (Fig. 2d). Although the *JASPer^cw2/cw2* mutant mostly phenocopies *JIL-1^z2/z2* mutant, polytene chromosomes retain their characteristic banded pattern (Fig. 2c, d), which are lost in the *JIL-1^z2/z2* mutant[30]. This observation is consistent with the partial lethality of *JASPer^cw2/cw2* mutant (54% of expected survival, *n* = 1496) as compared to the lethal *JIL-1^z2/z2* mutant (Fig. 2e). Ablation of JASPer by RNA interference in cultured cells also led to loss of JIL-1 (Fig. 2f). JIL-1

was depleted to the same level by RNAi against *jil-1* or *jasper* in S2 and Kc cells, suggesting that JIL-1 is unstable in the absence of JASPer. The *JIL-1* transcript level was unchanged upon *jasper* RNAi in our RNA-seq experiments, excluding regulation at the transcription level (Fig. 2g). However, trace amounts of JIL-1 or fragments of it might still be expressed and could explain the better viability of *JASPer^cw2/cw2* mutant versus *JIL-1^z2/z2* mutant[31].

**JASPer binds nucleic acids and H3K36me3 nucleosomes in vitro**. In addition to the LEDGF JIL-1-binding domain, JASPer harbors a PWWP domain at its N-terminus (Fig. 1c). PWWP domains have a positively charged surface favoring DNA binding and an aromatic pocket for methyl-lysine binding [for review[32]]. Conceivably, this domain is responsible for the recruitment of the JJ-complex to chromatin. As expected, recombinant JASPer had significant affinity for DNA in electrophoretic mobility shift assays (EMSA), while JIL-1 showed no detectable binding under the same conditions (Supplementary Fig. 4a). A 9-fold molar excess of JASPer shifted all DNA molecules. Apparently, several JASPer molecules can bind simultaneously one DNA molecule as at least three retarded bands appeared in the EMSA and the most retarded ones correlated with higher JASPer concentration. JASPer also bound a 123 nucleotide long RNA hairpin[33] in a dose-dependent manner (Supplementary Fig. 4b).

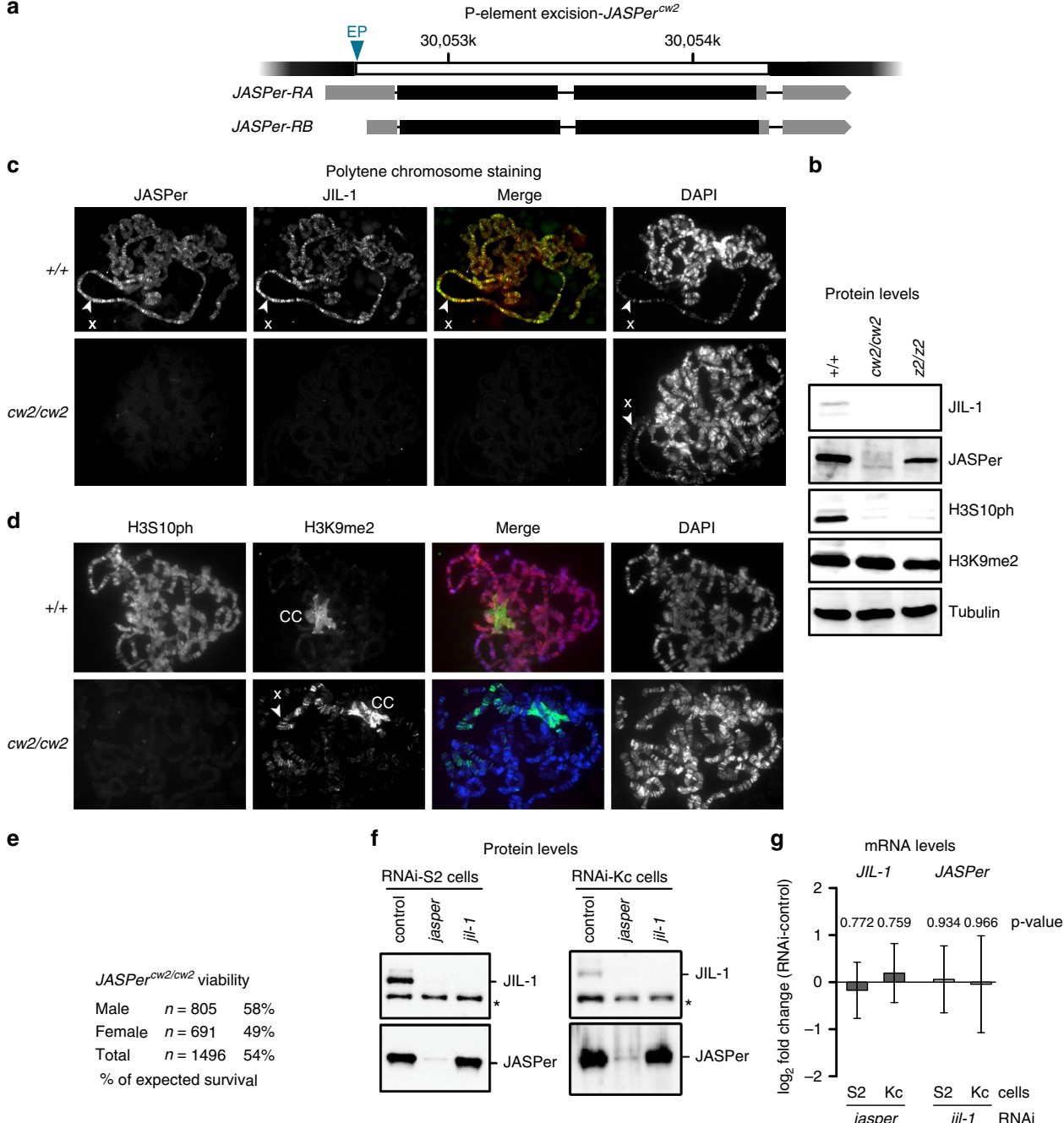

**Fig. 2** JIL-1 is unstable in absence of JASPer in the *JASPer*^cw2/cw2^ mutant and in cell lines. **a** Gene model for P-element excision in the *JASPer* locus to generate *JASPer*^cw2^ allele. The mRNA isoforms RA and RB are shown below. The excised genomic portion is marked in white and EP denotes the position of the excised P-element in EP-element line GS3268. **b** Western blot analysis of salivary gland extracts from L3 larvae of homozygous *JASPer*^cw2/cw2^ and *JIL-1*^z2/z2^ mutants and wild type larvae as control. Western blots using α-JIL-1, α-JASPer, α-H3S10ph, and α-H3K9me2 antibodies are shown, western blot with α-tubulin antibody was used as loading control. **c** Immunofluorescence microscopy of polytene chromosome squashes from L3 larvae of homozygous *JASPer*^cw2/cw2^ and wild type larvae as control. From left to right, staining for JASPer, JIL-1, merged images and DNA are shown. The X chromosome is marked by arrow heads. Source data are provided as a Source Data file. **d** Immunofluorescence microscopy of polytene chromosome spreads from L3 larvae of homozygous *JASPer*^cw2/cw2^ and wild type larvae as control. From left to right, staining for H3S10ph, H3K9me2, merged images and DNA are shown. The X chromosome is marked by arrow head and the chromocenter is labeled with "CC". **e** Table summarizing viability of male and female *JASPer*^cw2/cw2^ mutant flies. **f** Representative western blot analysis using α-JASPer and α-JIL-1 antibodies on whole cell extracts from S2 cells (left panel) and Kc cells (right panel) after *jasper* or *jil-1* RNAi treatment, as used for RNA-seq experiments. A cross-reacting band is marked by asterisk. Source data are provided as a Source Data file. **g** Bar chart showing mean log$_2$ fold-change of normalized mean RNA-seq counts for *JIL-1* and *JASPer* RNAi. Left panel, JIL-1 mRNA mean log$_2$ fold-change upon *jasper* RNAi (S2 $n = 4$ and Kc $n = 4$). Right panel, JASPer mRNA mean log$_2$ fold-change upon *jil-1* RNAi (S2 $n = 5$ and Kc $n = 4$). Error bars represent standard error of the mean.

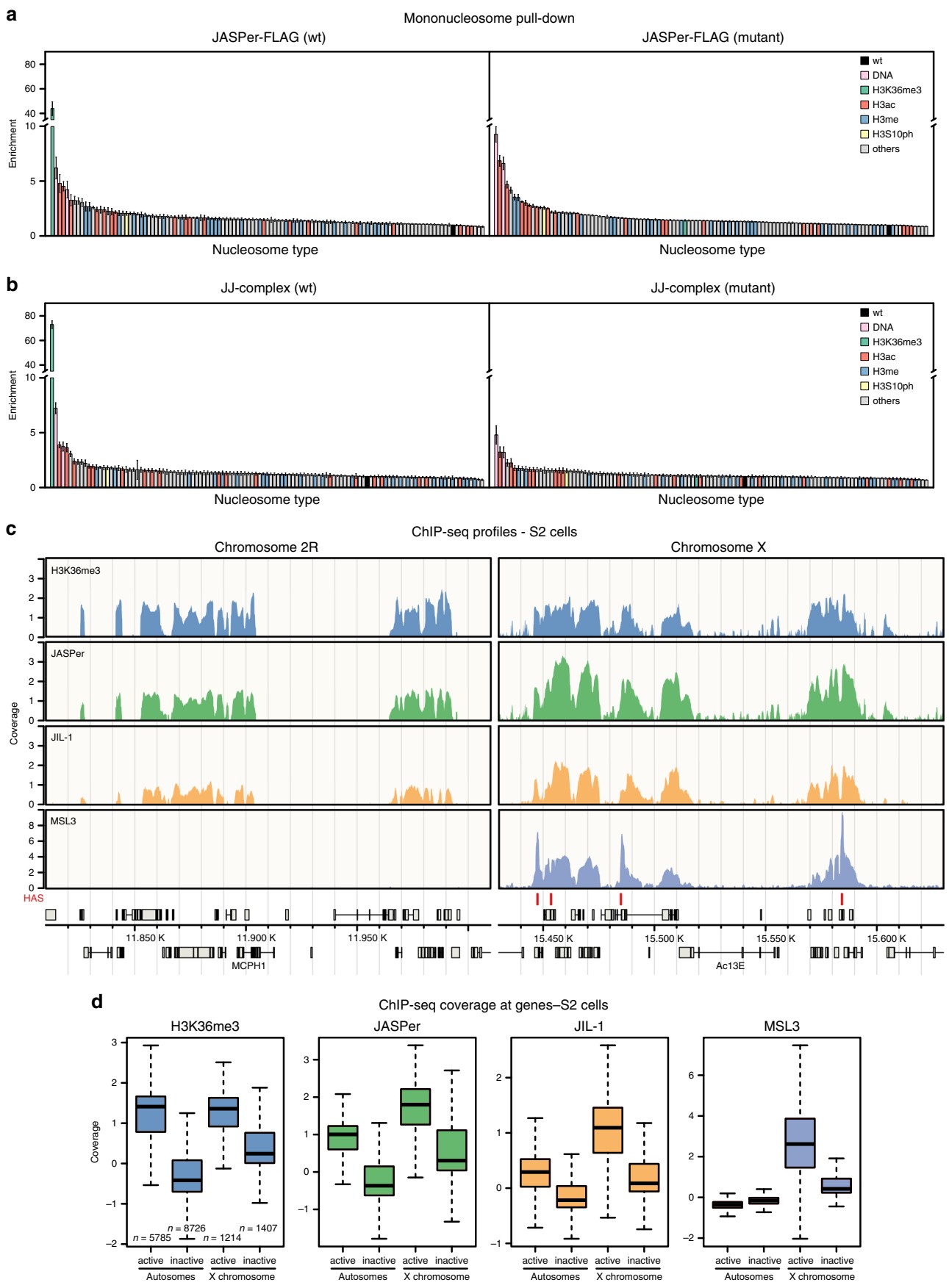

**Fig. 3** The JJ-complex binds H3K36me3 nucleosomes in vitro and in vivo, and is enriched on the male X chromosome. **a** Bar chart of mean enrichment ($n = 3$ independent experiments with 2 different protein preparations) of nucleosome library pull-down with JASPer-FLAG (left panel) and aromatic cage mutant (right panel) relative to unmodified nucleosome, which is set to 1. Error bars represent standard error of the mean. **b** Bar chart of mean enrichment ($n = 3$ independent experiments) of nucleosome library pull-down with JJ-complex (FLAG-JIL-1 and untagged JASPer) (left panel) and aromatic cage mutant (right panel) relative to unmodified nucleosome, which is set to 1, as in **a**. Error bars represent standard error of the mean. **c** Genome browser profile showing mean H3K36me3 (upper panel, $n = 4$ independent experiments), JASPer (second upper panel, $n = 4$ independent experiments with 2 different antibodies), JIL-1 (second lower panel, $n = 5$ independent experiments with 2 different antibodies) and MSL3 (lower panel, $n = 3$ independent experiments) MNase ChIP-seq normalized coverage along representative 200 kb windows on chromosome 2 R and X in male S2 cells. HAS are marked by red bars above the gene models in gray. **d** Box plot showing mean H3K36me3 (left panel, $n = 4$), JASPer (second left panel, $n = 4$), JIL-1 (second right, $n = 5$) and MSL3 (right, $n = 3$) MNase ChIP-seq normalized coverage, as in **c**, at active (tpm > 1) and inactive (tpm ≤ 1) genes on the autosomes ($n = 5785$ and $n = 8726$, respectively) and X chromosome ($n = 1214$ and $n = 1407$, respectively) in male S2 cells. Box plot elements are defined as center line marking the median, box limits are the upper and lower quartiles, whiskers extend maximally 1.5-times the interquartile range and outliers are removed.

To decipher the binding specificity of JASPer and the JJ-complex for nucleosomes, we used a library of 115 different types of DNA-barcoded nucleosomes bearing different histone and DNA modifications[34]. Recombinant, FLAG-tagged JASPer or FLAG-tagged JJ-complex were coupled to α-FLAG beads, incubated with the nucleosome library, washed and the pulled-down nucleosomes were quantified by sequencing of the associated indexes. Wild type JASPer showed high specificity towards nucleosomes bearing the single H3K36me3 modification (Fig. 3a). This modification was ~40-fold enriched in the IP relative to the unmodified nucleosome used for normalization. Mutation of two residues in the aromatic cage to alanines (Y23A and W26A) abolished specific H3K36me3 binding (Fig. 3a). Similar results were obtained for the JJ-complex, where H3K36me3 was ~70-fold enriched over the unmodified nucleosome, only if the aromatic cage is intact (Fig. 3b). In accordance with the DNA binding activity shown in EMSA (Supplementary Fig. 4a), we found a ~3-fold to 9-fold enrichment of the two nucleosome-free DNAs used as controls in the library. The enrichment (~2-fold to 7-fold) of nucleosomes bearing acetylated H3 tails, could be due to their lower assembly efficiencies, as described[34]. Alternatively, it could reflect, at least in part, the better accessibility of the linker DNA in those nucleosomes, as acetylation of the H3 tail decreases its binding to the linker DNA[35–37]. Acetylation may thus favor linker DNA-dependent binding by JASPer, as shown for the PWWP domain of PWWP2A[35]. These results point towards a mostly ionic interaction between the overall positively charged JASPer (pI of 8.3) and the negatively charged sugar-phosphate backbone of the DNA as has been proposed for other PWWP domains to synergize with the aromatic cage for high-affinity binding of H3K36me3 nucleosomes. The PWWP domain contacts both DNA gyres next to the H3 tail exit site through its basic surface and the aromatic cage engages with the K36me3 residue[38,39].

JIL-1 is a potent kinase in vitro and phosphorylates isolated H3 peptide (amino acids 1–20) and full-length histone H3. However, the isolated kinase proved to be inactive on nucleosome arrays in vitro even at high molar ratios of kinase to nucleosome[6]. To explore whether the oriented binding of the JJ-complex to nucleosomes would favor phosphorylation, we used a semi-quantitative kinase assay based on western blot detection of H3S10ph. Using H3K36me3-modified and unmodified nucleosomes and 12-mer nucleosome arrays (Supplementary Fig. 4c,d), we confirmed that only the wild type JJ-complex phosphorylates H3S10, that the active site-mutated enzyme is inactive and that we could detect low amounts of H3S10ph by western blot (Supplementary Fig. 4e). For the kinase assay with nucleosomes, we had to load ~10-times more of each reaction to detect similar levels of H3S10ph as compared to a completely phosphorylated, isolated H3, indicating that the JJ-complex is ~10-times less active on nucleosomes (Supplementary Fig. 4f). However, our analysis

showed that altogether JIL-1 in the JJ-complex is more active on nucleosome arrays (>3% of the phosphorylated H3 reference) than on mononucleosomes suggesting that the binding to one nucleosome in the array may facilitate the phosphorylation of a neighboring nucleosome (Supplementary Fig. 4f,g). The fact that we did not observe a preference for H3K36me3 nucleosomes is probably due to the high concentration of JJ-complex used in the kinase assay to allow H3S10ph detection by western blot.

**The JJ-complex localizes to active chromatin in vivo.** Because the JJ-complex specifically selects H3K36me3 nucleosomes via the PWWP of JASPer in vitro, we wished to confirm this interaction in vivo. Recently, differing results about JIL-1 localization in vivo arose from data generated using different ChIP-chip/-seq approaches[6,8,21], possibly due to different chromatin fragmentation protocols[40]. To clarify this issue, we used both chromatin digested with MNase and chromatin sheared by sonication for ChIPs of H3K36me3, JIL-1, JASPer, and MSL3 in male S2 cells and in female Kc cells (Supplementary Fig. 5a-c). Independent of the fragmentation strategy, we found that JIL-1 and JASPer binding profiles overlap with H3K36me3 at exons of active genes in vivo (Fig. 3c, Supplementary Fig. 6a-d), as expected[41]. Like JIL-1, JASPer is enriched at active genes on the X chromosome relative to autosomes only in male S2 cells (Fig. 3d, Supplementary Fig. 7a,b). We excluded that this enrichment is caused by normalization due to copy number differences by comparing the non-input-normalized coverages of H3K36me3, JASPer, and JIL-1 to the input (Supplementary Fig. 7c-e). In female Kc cells, we found similar coverages of H3K36me3 and JASPer at active genes on all chromosomes, whereas in male S2 cells the X chromosomal sequence coverage of H3K36me3 is roughly half of the autosomal one, as for the input. By contrast, coverages of JASPer and JIL-1 on active X chromosomal and autosomal genes in male S2 cells are similar. Interestingly, the X-chromosomal enrichment of JASPer and JIL-1 is only observed in male cells (Fig. 3d and Supplementary Fig. 7a).

The active genes on the X chromosome in male cells are strongly acetylated at H4K16 by the DCC subunit MOF, which is thought to decompact the chromatin fiber[42,43]. This loosening of chromatin folding may allow JASPer to bind better to H3K36me3, independent of JIL-1. To test whether the X-chromosomal enrichment of JASPer depends on JIL-1, we analyzed JASPer, MSL3, H4K16ac, and H3K9me2 distribution by ChIP-seq after jil-1 RNAi in S2 cells (Fig. 4a and Supplementary Fig. 5d). To quantify the absolute difference in ChIP-seq coverage between conditions by spike-in normalization, we added 5% *D. virilis* cells to our chromatin preparations[44]. Intriguingly, the X chromosome-specific enrichment of JASPer in male S2 cells was reduced to the autosomal level in absence of JIL-1, while the DCC subunit MSL3 was slightly redistributed, the diagnostic H4K16ac, set by the DCC, slightly dropped and the

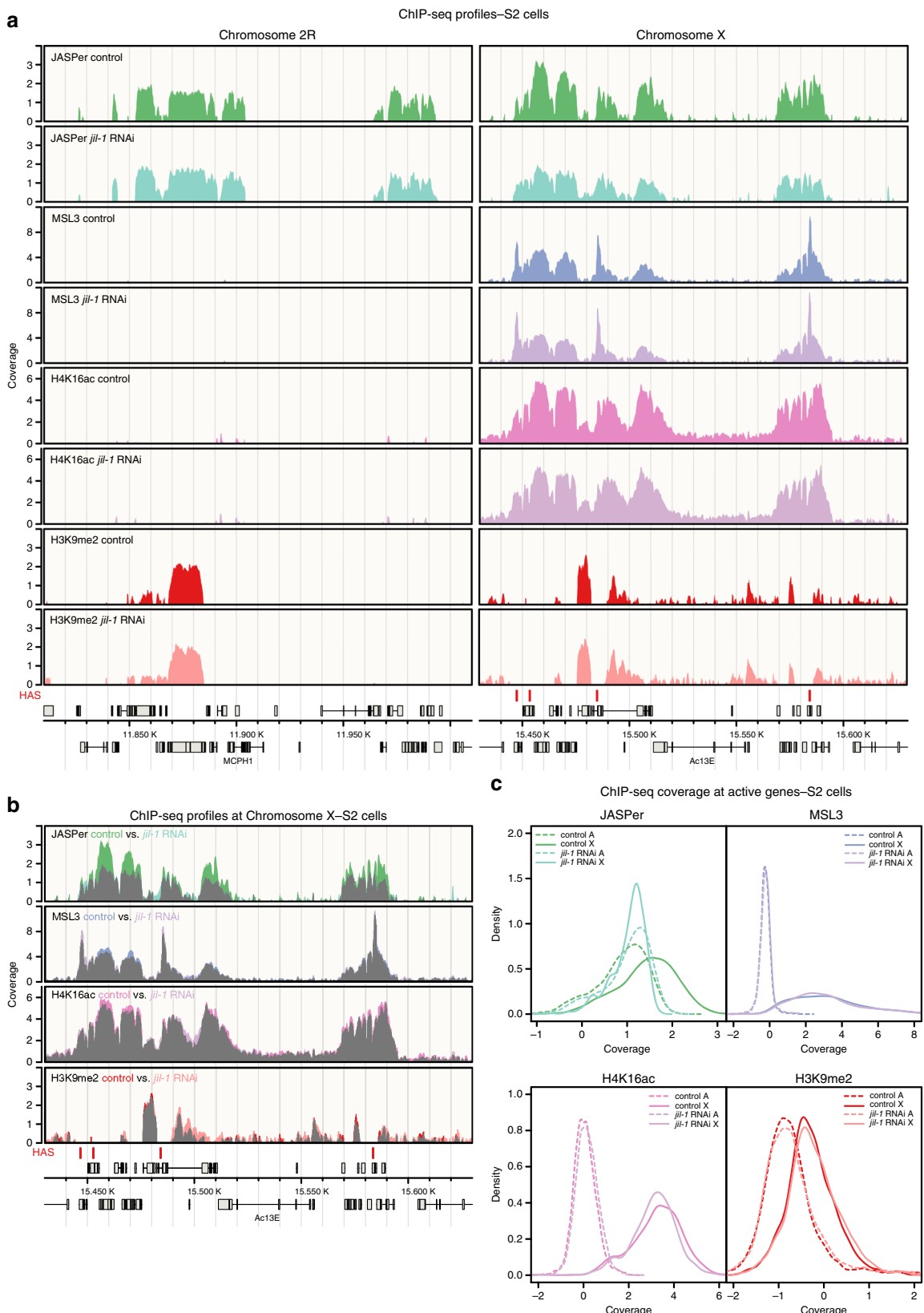

heterochromatin mark H3K9me2 slightly increased (Fig. 4 and Supplementary Fig. 8a,b). This demonstrates that JASPer per se does not need JIL-1 for H3K36me3 interaction, but its binding is enhanced on the male X chromosome in the JJ-complex. Interestingly, the loss of JASPer after depletion of JIL-1 is stronger closer to the ~300 high affinity sites (HAS) bound by the

DCC along the X chromosome (Supplementary Fig. 8c). Concomitantly, the spreading of MSL3 from HAS is slightly diminished and the H4K16ac density slightly drops but independently of the distance to HAS after *jil-1* RNAi. These small differences in the dosage compensation hallmark probably cannot explain the loss of JASPer enrichment. It thus appears that

**Fig. 4** JIL-1 and not H4K16ac is responsible for the enrichment of JASPer at the male X chromosome. **a** Genome browser profile showing mean ($n = 3$, for MSL3 $n = 2$) spike-in ChIP-seq normalized coverage in control male S2 cells and after *jil-1* RNAi treatment from top to bottom for JASPer, MSL3, H4K16ac and H3K9me2 along representative 200 kb windows on chromosome 2R and X. HAS are marked by red bars above the gene models in gray. **b** Genome browser profile as in **a** showing mean ($n = 3$, for MSL3 $n = 2$) spike-in ChIP-seq normalized coverage in control male S2 cells and after *jil-1* RNAi treatment, from top to bottom for JASPer, MSL3, H4K16ac, and H3K9me2 along a representative 200 kb window on chromosome X. Signal overlay is marked in grey. **c** Density plot showing mean ($n = 3$, for MSL3 $n = 2$) spike-in ChIP-seq normalized coverage in control male S2 cells and after *jil-1* RNAi treatment at active (tpm > 1) genes for JASPer (top left), MSL3 (top right), H4K16ac (bottom left), and H3K9me2 (bottom right). X chromosomal genes ($n = 1214$) are represented by a solid line and autosomal genes (chromosomes 2L, 2R, 3L, and 3R, $n = 5785$) by a dashed line.

JASPer's enrichment on the male X chromosome depends directly on JIL-1.

Because the main difference between the X chromosome and autosomes is the presence of the DCC and gene-body H4K16 acetylation, the enrichment of the JJ-complex on the X chromosome may be due to functional interactions of the JJ-complex with the DCC. Direct interaction of JIL-1 with MSL1 and MSL3 subunits of the DCC had been shown in vitro[45], but so far no clear direct association of the two endogenous complexes has been documented (see also below). We explored the interaction between the two recombinant complexes after expression from baculovirus vectors. Extracts containing JJ-complex (FLAG-JIL-1/untagged JASPer) on the one hand and a partial DCC consisting of MSL1, MSL2, and MSL3 (FLAG-MSL1/untagged MSL2/FLAG-MSL3) on the other hand were mixed in appropriate stoichiometry[46] and specific antibodies were used for IP (Supplementary Fig. 8d). The MSL1 antibody retrieved not only the associated MSL2 and MSL3, but also some JJ-complex. Conversely, the JIL-1 antibody immunoprecipitated MSL proteins in addition to abundant JJ-complex. This suggests that the two complexes may directly interact with each other. Altogether, the enrichment of the JJ-complex on the male X chromosome may be explained, at least in part, by a JIL-1-dependent interaction between the JJ-complex and the DCC.

**The JJ-complex supports expression of male X-linked genes**. As we confirmed that the JJ-complex binds to active gene bodies, we wished to explore the functional consequences. To do so, we quantified the transcriptome changes by RNA-seq after RNAi depletion of JASPer or JIL-1 in male S2 and female Kc cells. PCA analysis showed that *jasper* and *jil-1* RNAi affected overall gene expression similarly (Supplementary Fig. 9a). The per-gene analysis showed that upon *jasper* and *jil-1* RNAi in both cell lines the transcription of many genes changed over a wide range of expression levels, with more genes being downregulated (fdr < 0.05) (Fig. 5a). The changes upon *jasper* and *jil-1* RNAi correlate ($r = 0.597$ in S2 and $r = 0.561$ in Kc cells), indicating that depletion of the JJ-complex and of JIL-1 alone result in a similar phenotype (Supplementary Fig. 9b). Remarkably, transcription of X chromosomal genes is globally reduced upon depletion of either protein in male S2, but not in female Kc cells (Fig. 5b). We showed earlier that mapping JIL-1-dependent interphase H3S10ph in exponentially growing cells is not possible due to the overwhelming levels of mitotic H3S10ph[6]. Instead, we monitored changes in the diagnostic histone modifications H3K9me2 for heterochromatin and H4K16ac for dosage compensation upon JIL-1 depletion in male S2 cells using the spike-in ChIP-seq approach (Supplementary Fig. 9c). In agreement with a decreased expression of X-chromosomal genes, a small increase of H3K9me2 and a slight decrease of H4K16ac were observed. In a gene-by-gene analysis of X-linked genes, we observed that downregulated genes consistently tend to lose H4K16ac and JASPer or gain H3K9me2 (Supplementary Fig. 10). We related the slight increase of H3K9me2 to an increased susceptibility of

the X chromosome to invasion of patches of heterochromatin as also seen in control cells (Supplementary Fig. 9c). The decrease in H4K16ac on the male X chromosome observed both globally (Fig. 4a,b) and specifically at expressed genes (Supplementary Fig. 9c) suggests that JIL-1 may affect H4K16ac indirectly.

Altogether our results suggest that JIL-1 overall positively regulates gene expression and that the effect is most pronounced on the X chromosome in male cells.

**The JJ-complex regulates expression of telomeric transposons**. JIL-1 is the only known activator of the expression of non-LTR retrotransposons of the HTT arrays (*HeT-A, TAHRE,* and *TART-A/B/C*), which is essential for telomere maintenance in *Drosophila*[18,19]. Mapping our ChIP-seq data to the consensus sequences of 126 *D. melanogaster* transposable elements (TEs) we found that a subset of them showed an enrichment of H3K36me3 and JJ-complex in S2 cells (Fig. 6a). H3K36me3, JASPer, and JIL-1 are strongly enriched at all transposons of the HTT arrays, as well as at the LTR-retrotransposons *Gypsy5* and *3S18* (Fig. 6a). Depletion of JIL-1 and JASPer by RNAi led to statistically significant reduced expression (fdr < 0.05) of the majority of TEs (Fig. 6b, Supplementary Fig. 11a). The good correlation of the effects of each RNAi ($r = 0.850$) supports the idea of a joint action of JIL-1 and JASPer in a functional complex (Supplementary Fig. 11b). Among telomeric TEs, which are bound by the JJ-complex, the expression of *HeT-A* and *TART-A* is reduced after JASPer depletion and *TART-B* and *TART-C* are additionally downregulated after JIL-1 depletion. However, we do not robustly detect expression of *TAHRE*. Even though we found many more significantly downregulated TEs in S2 cells, we propose that this is indirect as these TEs lack detectable H3K36me3 enrichment and JJ-complex binding (Fig. 6a, b). However, the TEs of the HTT arrays seem to be mostly active and lack H3K9me2. Upon JIL-1 depletion, we detected an increase in H3K9me2 at the TEs of the HTT arrays, except for *TART-C* (Fig. 6c, Supplementary Fig. 11). Concomitantly, the enrichment of JASPer is decreased at all transposons of the HTT arrays upon JIL-1 depletion (Supplementary Fig. 12), suggesting that either H3K36me3 is decreased there because of the lower expression and/or JIL-1 contributes to the enrichment of the JJ-complex at telomeres.

Altogether, we propose that TE's of the HTT arrays acquire H3K36me3 when they are transcribed and recruit the JJ-complex to maintain their active state at least in part by preventing heterochromatization.

**The JJ-complex associates with other chromatin complexes**. To elucidate the interaction network of the JJ-complex, we immunoprecipitated JASPer with various antibodies under stringent conditions from embryo extracts and identified associated proteins by mass spectrometry. We identified 69 statistical significantly enriched proteins (*p*-value < 0.05 and log$_2$ fold-change > 4) (Fig. 7a, Supplementary Data 1). The five most enriched GO terms associated to those proteins include 'chromatin remodeling', 'protein acetylation', 'chromatin organization', and

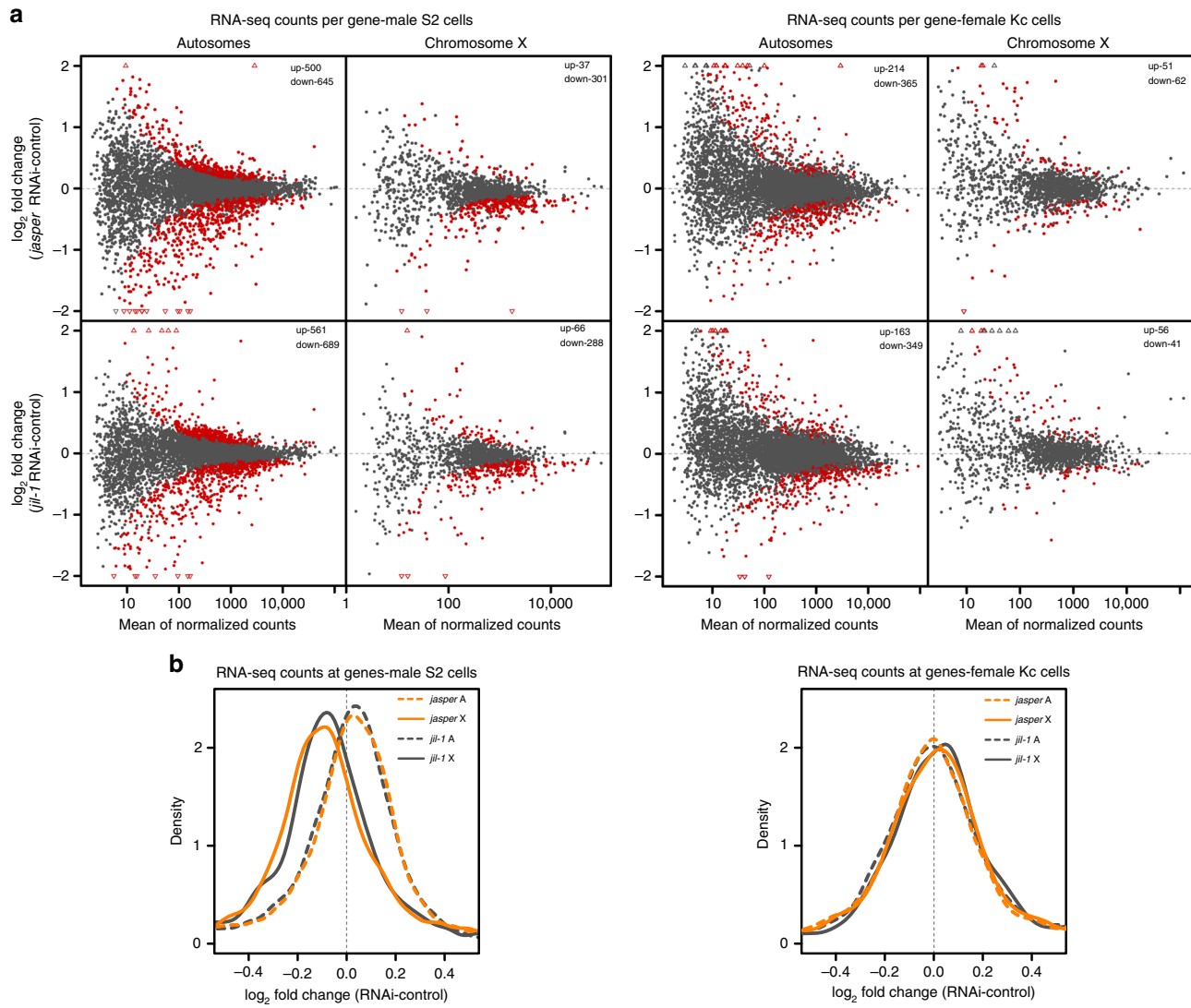

**Fig. 5** JIL-1 and JASPer depletion in cells modulates the transcriptional output of genes, especially on the male X chromosome. **a** MA-plot showing mean log₂ fold-change of RNA-seq counts upon *jasper* RNAi versus control (upper panel, $n = 4$) and *jil-1* RNAi versus controls (lower panel, $n = 5$) against mean RNA-seq counts for robustly detected genes at autosomes (left, chromosomes 2L, 2R, 3L, and 3R $n = 6833$) and X chromosome (right, $n = 1441$) in male S2 cells (left site). Statistically significant differentially expressed genes between RNAi and control conditions (fdr < 0.05) are marked in red and the number of significant genes is indicated on the plot. On the right, mean log₂ fold-change of RNA-seq counts upon *jasper* RNAi versus control (upper panel, $n = 4$) and *jil-1* RNAi versus controls (lower panel, $n = 4$) against mean RNA-seq counts for autosomal genes (left, chromosomes 2L, 2R, 3L, and 3R $n = 7144$) and X chromosomal genes (right, $n = 1509$) in female Kc cells (left site). **b** Density plot showing mean log₂ fold-change of RNA-seq counts upon *jasper* RNAi versus controls ($n = 4$) and *jil-1* RNAi versus controls ($n = 5$) at genes in male S2 cells, in left panel, as in **a**. X chromosomal genes ($n = 1441$) are marked with solid line and autosomal genes (chromosomes 2L, 2R, 3L, and 3R, $n = 6833$) with dashed line and *jasper* RNAi additionally in orange. Right panel, mean log₂ fold-change of RNA-seq counts upon *jasper* RNAi and *jil-1* RNAi versus controls ($n = 4$ each) at genes in female Kc cells. X chromosomal genes ($n = 1509$) and autosomal genes (chromosomes 2L, 2R, 3L, and 3R, $n = 7144$).

'transcription from RNA Pol II promoters' and its regulation (Fig. 7b). Among the most enriched interacting proteins we found BOD1, Dpy-30L1, Rbbp5, and Set1, subunits of the Set1/ COMPASS complex mediating promoter-proximal H3K4 dimethylation and trimethylation [for review[47]]. Dpy-30L1 and Rbbp5 are common subunits of the different COMPASS complexes, containing one of the three histone methyltransferases Set1, Trx, and Trl in flies. Interestingly, BOD1/CG5514 had not been described in the *D. melanogaster* Set1/COMPASS complex but is a specific subunit of the Set1B/COMPASS complex in humans[48,49]. The next most represented interactors were the related PBAP and Brm remodeling complexes with e(y)3, poly-bromo, Bap170, Bap111, and Snr1 (Fig. 7a and Supplementary Fig. 13). Further subunits of the PBAP/Brm complex and other

subunits of remodeling complexes were also enriched, though below statistical significance of this experiment (Fig. 7b). Furthermore, we found the heterochromatin components Su(var)3-7 and Su(var)205 (HP1) significantly enriched (Fig. 7a), which are known to genetically interact with JIL-1[14,15]. Several published interactors of JIL-1, like Chromator[50] or MSL1 and MSL3[45] were not detected or not significantly enriched, possibly because of more dynamic association. Among the subunits of the DCC, only MOF was detected together with other subunits of the alternative MOF-containing NSL (non-specific-lethal) complex (Fig. 7a). NDF (nucleosome destabilizing factor) which was found associated with JIL-1 by mass spectrometry after cross-linking[51] was also enriched (Fig. 7a). NDF has recently been shown to destabilize nucleosomes in front of the transcribing polymerase, but

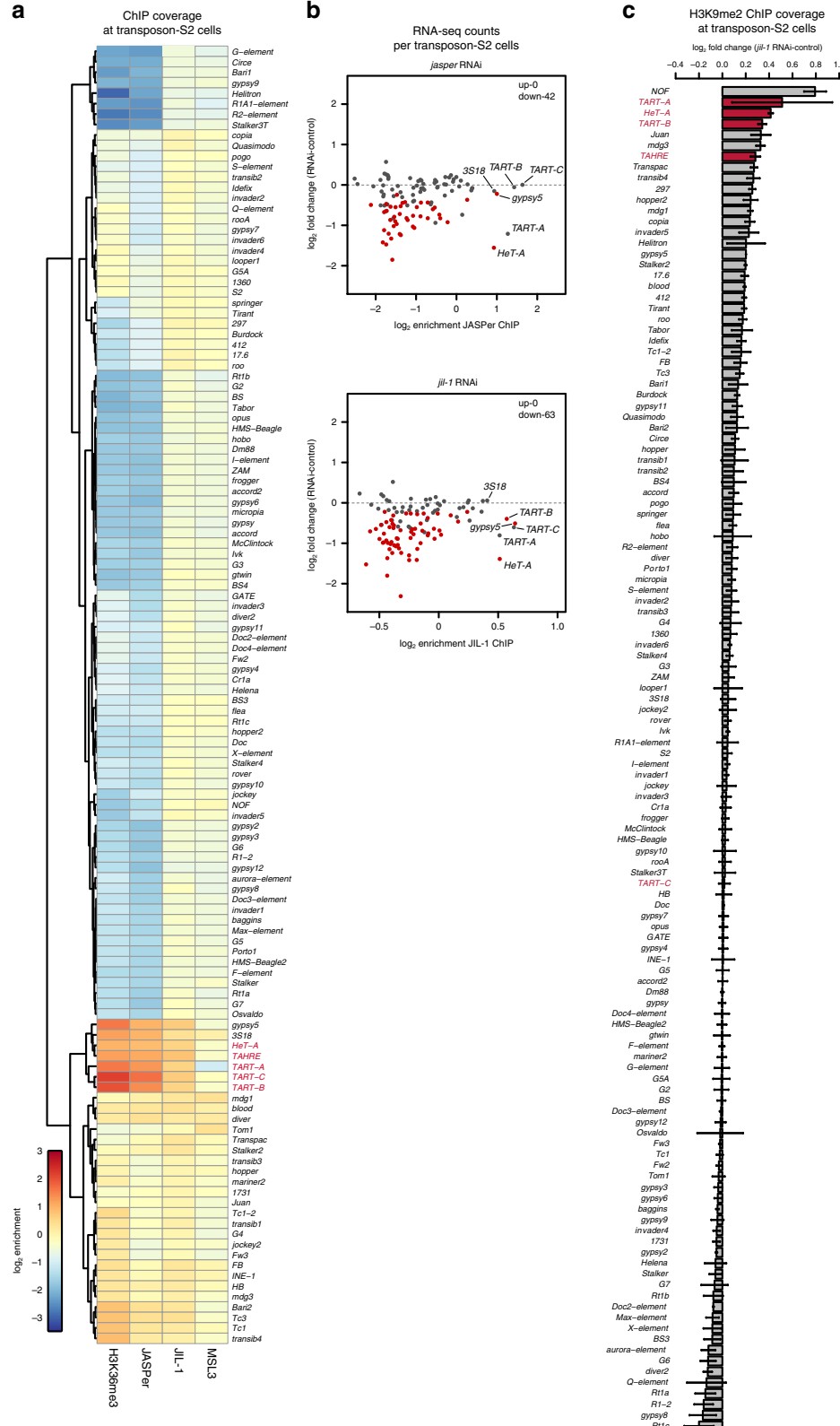

its depletion had only minor effects on overall transcript levels[52]. We speculated that the JJ-complex and NDF may have redundant functions on transcription. Therefore, we compared the transcriptome changes in male S2 cells after RNAi depletion of either JASPer or NDF alone, or in combination. Although, the depletion efficiency is only partial for NDF (Supplementary Fig. 13a), PCA separates the single *jasper* and *ndf* RNAi samples well from the control samples (Supplementary Fig. 13b). However, the combined depletions showed no increased variance, and the expression changes after JASPer or NDF depletion show only a weak correlation ($r = 0.39$, Supplementary Fig. 13c). Although, JASPer and NDF co-localize to active genes marked by H3K36me3, they seem to not have redundant roles in the regulation of steady state mRNA levels.

**Fig. 6** JIL-1 and JASPer depletion in S2 cells decrease the transcript level of transposons of the telomeric transposons of the HTT arrays. **a** Heatmap showing mean normalized $\log_2$ enrichment in H3K36me3 ($n = 4$), JASPer ($n = 4$), JIL-1 ($n = 5$), and MSL3 ($n = 3$) MNase ChIP-seq at transposons ($n = 124$) in male S2 cells. Transposons of the HTT array are marked in red. **b** Scatter plot showing mean $\log_2$ fold-change of RNA-seq counts upon *jasper* RNAi versus control and *jil-1* RNAi versus control against mean normalized $\log_2$ enrichment in JASPer ($n = 4$, upper panel) and JIL-1 ($n = 5$, lower panel) MNase ChIP-seq, respectively, at robustly detected transposons in male S2 cells ($n = 111$). Statistically significant differentially expressed transposons between RNAi and control conditions (fdr < 0.05) are marked in red and the number of significant genes is indicated on the plot. TEs of the HTT arrays, *gypsy5*, and *3S18* are labeled. **c** Bar plot of difference of mean H3K9me2 ($n = 3$ each) spike-in ChIP-seq normalized coverage after *jil-1* RNAi treatment and control male S2 cells at transposons ($n = 124$) in male S2 cells. Error bars represent standard error of the mean. TEs of the HTT arrays are marked in red.

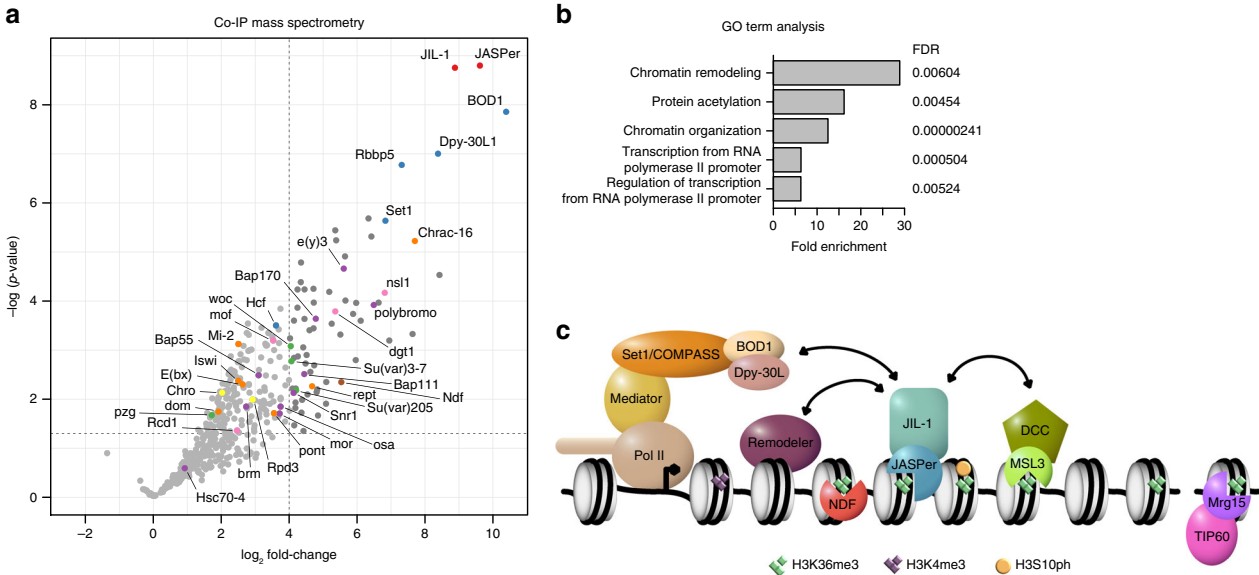

**Fig. 7** The JASPer interaction network and other H3K36me3 binding proteins and complexes in *Drosophila melanogaster*. **a** Volcano plot of IP-MS showing $-\log_{10}(p\text{-values})$ against mean $\log_2$ fold-change in α-JASPer IP ($n = 6$) versus control IP ($n = 5$). Significantly enriched ($p$-value < 0.05 and $\log_2$ fold-change > 4) proteins ($n = 69$) are highlighted in dark gray. JIL-1 and JASPer are marked in red, Set1/COMPASS complex members in blue, PBAP/Brm complex members in purple, other proteins involved in chromatin remodeling in orange, NSL complex members in pink, Su(var)3-7, Su(var)205, woc and pzg in green, Chro and Rpd3 in yellow and NDF in brown. **b** Bar plot showing GO term enrichment of significantly enriched proteins shown in **a**. The five statistically significantly (fdr < 0.01) most enriched GO terms are shown. **c** Model of JJ-complex binding at H3K36me3 marked gene bodies and interactions with other complexes. Interactions presented here are indicated by arrows. Other known H3K36me3 binding proteins (NDF and Mrg15) are drawn at the lower side. We propose that phosphorylation by JIL-1 kinase is tightly regulated in space and time in part by its partner JASPer which stabilizes and anchors the kinase to active genes and telomeric transposons by binding to H3K36me3 nucleosomes via its PWWP domain. Regulation by JASPer affects any potential phosphorylation by JIL-1, in particular H3Ser10 phosphorylation which is involved in inhibition of heterochromatinisation.

In summary, we found that the JJ-complex associated with Set1/COMPASS and several nucleosome remodeling complexes. These interactions provide links for understanding the regulation of chromatin structure and function through the JJ-complex.

## Discussion

We showed that JIL-1 kinase forms as stable complex with a so far uncharacterized protein encoded by *CG7946*. We named the protein JASPer (JIL-1 Anchoring and Stabilizing Protein). Together the proteins form the JASPer/JIL-1 (JJ)-complex (Fig. 1), which is the major form of JIL-1 kinase in vivo, since JIL-1 is unstable in the absence of JASPer (Fig. 2). The interaction is mediated by a short stretch of conserved residues within JIL-1's CTD containing a conserved FxGF motif and the LEDGF domain of JASPer. This interaction mode seems to be conserved throughout the animal kingdom, since the human JASPer ortholog PSIP1 (or LEDGF/p75) binds via its LEDGF/IBD (Integrase Binding Domain) domain to various interaction partners, including HIV integrase, MLL1-MENIN complex and IWS1 containing the conserved FxGF motif[29]. These interactions may also trigger deleterious targeting. For example, PSIP1 is hijacked

by the HIV integrase to ensure integration of the viral genome in active chromatin, or PSIP1 mis-targets the MLL1 fusion in mixed-lineage leukemia (MLL), inducing malignant transformation. Interestingly, the stability of the interaction with MLL1 is regulated through phosphorylation[29]. We found similar proteins and complexes associated with the JJ-complex under stringent IP-MS conditions. The most prominent interactors, Dpy-30L1, BOD1, Rbbp5, and Set1 are subunits of the Set1/COMPASS complex, which is related to the human MLL complexes. Several subunits of the PBAP/Brm complex, as well as other remodeling complexes are also enriched with the JJ-complex and contribute to the most enriched GO term (Fig. 7).

We suggest that JASPer drives the targeting of JIL-1 to active chromatin through its PWWP domain. The protein binds DNA and RNA, as well as H3K36me3 nucleosomes in vitro. We propose that the recruitment of the JJ-complex to the body of active genes enriched in H3K36me3 (Fig. 3) is the main recruitment mode of JIL-1 kinase to chromatin, but we do not exclude that additional binding modes are relevant at promoters and enhancers as described earlier[8,21]. Those binding modes could implicate interactions with other chromatin complexes, RNA or DNA. Recently, the protein PWWP2A protein was described to bind

H2A.Z-containing nucleosomes at the 5' end of transcribed genes, as well as active gene bodies decorated with H3K36me3 using two different binding modules[35,53].

The same targeting principle by JASPer binding via its PWWP domain to H3K36me3 may be used to recruit JIL-1 to telomeric HTT transposons (Fig. 6). However, it is not clear if those transposons acquire H3K36me3 through the Set2-dependent methylation associated with elongating RNA Pol II, as coding genes do[54] or by another mechanism.

The recombinant JJ-complex has a strong kinase activity towards S10 on isolated H3 in vitro but the efficiency of phosphorylating H3S10 in nucleosomes is very low (Supplementary Fig. 4). H3K36me3 is essential to bring JIL-1 to active chromatin, but is not sufficient to unleash its kinase activity upon nucleosomes in vitro. We speculate that JIL-1 may need to be activated by specific signals generated within chromatin or downstream of a signaling pathway, similarly to its orthologous kinases MSK1/2 [for review see ref. 55]. However, the nucleosome may not be the physiological substrate for JIL-1. During the course of transcription, nucleosomes are disassembled and evicted histones associate with various chaperones to be reassembled after the passage of the polymerase [for review see ref. 56]. Thus, H3 substrate for JIL-1 phosphorylation in vivo could also be any intermediate, occurring during the transcription process.

Methylation of H3K36 at active chromatin has pleiotropic functions in various model organisms, such as repression of spurious transcription, alternative splicing, DNA repair and recombination [for review see ref. 57]. We summarize in Fig. 7c, the different factors known to localize to H3K36me3 chromatin in *D. melanogaster*. Recently, a H3K36R mutant of the replication-dependent H3 in *D. melanogaster*, resulting in almost complete loss of H3K36me3, showed that this residue is essential for viability[58] and triggers dysregulation of transcript levels mostly by post-transcriptional mechanisms[59]. Our results are compatible with an indirect function of H3K36me3 on transcriptional output.

It is attractive to speculate that JIL-1 may affect gene activity indirectly through installation of a phospho-methyl switch in interphase. Accordingly, phosphorylation of H3S10 prevents methylation of H3K9, which would hinder heterochromatinization by inhibiting further H3K9 methylation and HP1 binding[9,14]. We found that in *JASPer^cw2/cw2* deficient flies, as already described for *JIL-1^z2/z2* null flies, heterochromatin histone marks and HP1 spread from the chromocenter especially to the X chromosome on polytene nuclei of salivary glands (Fig. 2d). JIL-1 depletion in S2 cells induces a small but significant increase of H3K9me2 on genes (Supplementary Figs. 8, 9c) in S2 cells. The quantification of H3K9me2 on transposons of the X chromosome is not yet possible due to the lack of annotation of these elements. However, transposons of the HTT arrays, which are in a heterochromatic environment, clearly acquire H3K9me2. The apparent difference in the magnitude of H3K9me2 spreading between the salivary gland cells and S2 cells might be due to several reasons. First, the strong mitotic H3S10ph by Aurora B kinase might reset the system at each cell division in cultured cells. Second, the endoreplication in salivary gland cells might exacerbate the antagonism between H3S10ph and H3K9me2 due to replication as described in mESCs[7]. Third, the absolute amount of H3K9me2 spreading on the X chromosome, although evident on polytene chromosome preparations might be low, as observed in mESCs[7].

The most prominent effect in our RNA-seq experiments is the specific reduction of X-chromosomal transcription in male S2 but not female Kc cells. Because the JJ-complex is also enriched on the X chromosome in male cells in a JIL-1- and dosage compensation dependent manner (Fig. 4), there may be a link either

to the specific compensation mechanism established by the DCC or to a more general compensation mechanism known to occur in response to variation in copy number of genes. Such a 'generic' compensation mechanism has been described in many eukaryotes, as well as in various *Drosophila* cell lines[60,61]. There are two main arguments for the first scenario: the enrichment of the JJ-complex on the X chromosome in males depends on the DCC and decreases with increasing distance to HAS [ref. 6 and Supplementary Fig. 8b] and we documented a weak but consistent interaction of the recombinant JJ-complex and partial DCC (Supplementary Fig. 8c).

Finally, we can imagine a role for the JJ-complex in the context of safeguarding genome stability threatened by R-loop formation. The presence of H3S10ph at transcribed regions has been related to the formation of R-loops, and proposed to be important to prevent genomic instability[62,63]. In *Drosophila* cells, almost 50% of R-loops detected by GRID-seq appear within genes[64]. The formation of R-loops and associated proteins could trigger the activation of JJ-complex for efficient H3S10 phosphorylation. Ectopic H3S10ph by JIL-1 correlates with large-scale chromatin opening in vivo[65] although H3S10ph per se has no effect on chromatin structure in vitro[66]. Conceivably, other proteins than H3 may be relevant substrates for the JJ-complex. Chromatin remodeling enzymes, which correspond to the most significantly overrepresented GO term in our unbiased IP-MS analysis of the JJ-complex interaction network would be good candidates for such regulation.

In summary, we showed that JASPer is essential for JIL-1 function: it stabilizes JIL-1 and recruits it to transcribed chromatin. Future goals will be to unravel the signaling events that lead to activation of the JJ-complex, its non-histone substrates and role in modulating chromatin structure and function.

## Methods

**Cell culture and RNAi**. S2-DRSC (DGRC stock # 181), Kc167 (DGRC stock # 1) cells were cultured in Schneider's *Drosophila* Medium (Thermo Fisher), supplemented with 10% heat-inactivated Fetal Bovine Serum (Sigma-Aldrich), 100 units/mL penicillin and 0.1 mg/mL streptomycin (Sigma-Aldrich) at 26 °C. RNAi against target genes in S2 and Kc cells for ChIP-seq was performed for 7 days in 1 or 2 flasks (75 cm²) seeded with 12 million cells and treated with 50 μg dsRNA/flask after a wash in serum free medium[46]. Fresh medium was added at day 5 to sustain growth. For RNAi against target genes in S2 and Kc cells for RNA-seq, cells were washed with serum-free medium and 10 μg dsRNA per 10⁶ cells at a concentration of 10 μg/mL in serum-free medium (10⁶ cells in 6-well plate) was added, incubated for 10 min at room temperature (RT) with slight agitation and further 50 min at 26 °C. Two volumes of complete growth medium were added and cells were incubated for 3 days at 26 °C. At day 3, cells were split, reseeded and retreated as at day 1. Cells were incubated for further 4 days at 26 °C. dsRNA was generated from PCR products obtained using the following forward and reverse primers (separated by comma):
*jasper* RNAi #1: TTAATACGACTCACTATAGGGAGAATGGGTAAGGAA, TTAATACGACTCACTATAGGGAGAGGAGGTGCTAGT;
*jasper* RNAi #2: TTAATACGACTCACTATAGGGAGATGGAGAACGCCCGCAAAGAA, TTAATACGACTCACTATAGGGAGATTGCCCACATACCGGCGAAG;
*jil-1* RNAi #1: TTAATACGACTCACTATAGGGAGACAGCAGCGTCG, TTAATACGACTCACTATAGGGAGATTGGAACTGAT;
*jil-1* RNAi #2: TTAATACGACTCACTATAGGGAGACAGTGGTTATCCCTTCGCA, TTAATACGACTCACTATAGGGAGATACCGCGGAGAATGAATACC;
*gst* RNAi: TTAATACGACTCACTATAGGGAGAATGTCCCCTATACTAGGTTA, TTAATACGACTCACTATAGGGAGAACGCATCCAGGCACATTG;
*gfp* RNAi: TTAATACGACTCACTATAGGGTGCTCAGGTAGTGGTTGTCG, TTAATACGACTCACTATAGGGCCTGAAGTTCATCTGCACCA;
*ndf* RNAi #1: TTAATACGACTCACTATAGGGAGAATCGGTCAAGTCGACAAAGG, TTAATACGACTCACTATAGGGAGATCATTCCAAGACCCAGGAAGC;
*ndf* RNAi #2: TTAATACGACTCACTATAGGGAGACCGAAAGCAAAGTCCGTGG, TTAATACGACTCACTATAGGGAGAAACCTTGTGACCCGTGTAGA;
*D. virilis* 79f7Dv3 cells[67] (kind gift of B. V. Andrianov) were cultured in Schneider's *Drosophila* Medium (Thermo Fisher), supplemented with 5% heat-

inactivated Fetal Bovine Serum (Sigma-Aldrich), 100 units/mL penicillin, and 0.1 mg/mL streptomycin (Sigma-Aldrich) at 26 °C.

Sf21 cells (Thermo Fischer) were cultured in SF900 II SFM (Thermo Fisher), supplemented with 10% heat-inactivated Fetal Bovine Serum (Sigma-Aldrich), 0.1 mg/mL gentamicin (Sigma-Aldrich) at 26 °C.

**Recombinant protein expression and purification.** For purification of GST-JASPer fusion protein, the coding sequence of *JASPer* (*CG7946-RA*) from EST clone LD23804 was cloned into pGEX-4T2. GST-JASPer was expressed in *E. coli* Rosetta 2 (DE3) (Merck) and purified using Glutathione Sepharose High Performance beads (GE Healthacare) for antibody generation. For all biochemical assays, we used the baculovirus expression system in Sf21 cells. For purification of recombinant JASPer and aromatic cage mutant (Y23A and W26A) by FLAG-tag affinity chromatography, the coding sequence of *JASPer* was directly fused to a C-terminal coding sequence of FLAG affinity tag and cloned into pFBDM under control of the polyhedrin promoter[22]. For dual expression of the JJ-complex, we cloned *FLAG-JIL-1* or active sites mutant (D407A and D759A) and fragments thereof into pFBDM under the control of the polyhedrin promoter together with untagged *JASPer* or aromatic cage mutant (Y23A and W26A) and fragments thereof under the control of the p10 promoter. An N-terminal FLAG tag was directly cloned in front of the *JIL-1* gene (JIL-1-RA)[16]. FLAG-MSL3 was expressed from pFastBac1 as described[68]. FLAG-MSL1 was expressed from pFBDM under the control of the polyhedrin promoter together with untagged MSL2 under the control of the p10 promoter (Müller et al., manuscript in preparation).

The JJ-complex and JASPer were expressed in Sf21 cells and purified by FLAG-tag affinity chromatography, as previously described[69] with minor modifications. In brief, Sf21 cells at $10^6$ cells/mL ($250 \times 10^6$ cells) were infected 1:1000 (v/v) with baculovirus, expressing JJ-complex or JASPer-FLAG. After 72 h, cells were harvested and washed once in Phosphate buffered saline (PBS), frozen in liquid nitrogen and stored at −80 °C. To lyse, cells were rapidly thawed, resuspended in 5 mL Lysis Buffer per 50 mL of culture (50 mM HEPES pH 7.6, 400 mM NaCl, 1 mM MgCl₂, 5% (v/v) glycerol, 0.5% (v/v) IGEPAL CA-360, 1 mM DTT) supplemented with cOmplete EDTA-free Protease Inhibitor Cocktail (Sigma-Aldrich) (PI). The suspension was sonicated for 60 s at 20% Amplitude (Branson-Sonifier) with 5 s 'on' and 10 s 'pause' cycles. Cell extract was treated with 1 μL Benzonase (Merck), supplemented with 0.15% (v/v) Triton-X-100, incubated with end-over-end rotation for 30 min at 4 °C and spun down at 4 °C for 30 min at 50,000×g. The supernatant was used for FLAG-tag affinity purification with 1.2 μL of FLAG-M2 bead bed volume (Sigma-Aldrich) per 1 mL of culture. Beads were first washed thrice in 20 bed volumes of Lysis Buffer, subsequently supernatant was added and incubated with end-over-end rotation for 3 h at 4 °C. Beads were pelleted (4 °C, 5 min, 500×g) and supernatant was removed. Beads were washed twice with 20 bed volumes each of Lysis Buffer, Wash Buffer (Lysis Buffer with 1 M NaCl) and finally twice with 20 bed volumes Elution Buffer (Lysis Buffer with 200 mM NaCl). For protein elution, beads were incubated with 0.2 bed volumes of Elution Buffer containing 5 mg/mL FLAG peptide (Sigma-Aldrich) for 10 min at 4 °C and subsequently 0.6 bed volumes of Elution Buffer with PI were added and incubated with end-over-end rotation for 2 h at 4 °C. The elution step was repeated and elution fractions were combined and concentrated if needed. Protein concentration was determined using BSA standards on SDS-PAGE with Coomassie brilliant blue G250 staining. Protein samples were flash-frozen in aliquots in liquid nitrogen and stored at −80 °C. For nucleosome IP, buffer was exchanged by adding 9 volumes of Exchange Buffer 1 (50 mM HEPES pH 7.6, 500 mM NaCl, 1 mM MgCl₂, 10% (v/v) glycerol, 0.05% (v/v) Triton-X-100, 1 mM DTT, 0.5 mM EDTA, 2.5 mM L-Aspartate), and concentrating with 30 MWCO Amicon Ultra-15 (Merck) to the starting volume. The proteins were again diluted in 9 volumes of Exchange Buffer 2 (Exchange Buffer 1 with 200 mM NaCl) and concentrated with 30 MWCO Amicon Ultra-15 (Merck).

**Electro mobility shift assay.** EMSA with dsDNA was performed as described in ref. [70], with slight modifications. In brief, binding reactions containing 70 nM 40 bp Cy5-labeled dsDNA (CCTGGA-GAATCCCGGTGCCGAGGCCGCTCAATTGGTCGTA) in Binding buffer (50 mM HEPES pH 7.6, 50 mM NaCl, 10% (v/v) Glycerol, 2 mM MgCl₂, 10% (w/v) BSA) were incubated for 10 min at RT. EMSA with RNA was performed as described in ref. [33] with 2.5 nM 123 nt $^{32}$P-labeled roX2-123 RNA in EMSA buffer (25 mM HEPES pH 7.6, 100 mM KCl, 3 mM MgCl₂, 1 mM DTT, 5% (v/v) glycerol, 100 μg/mL yeast tRNA (Sigma)) for 15 min at 20 °C. The protein:DNA/RNA complexes were resolved by native PAGE (4% gel in 0.5× TBE running buffer).

**Generation of *JASPer* null mutant fly line.** The *JASPer* null allele cw2 was isolated in a screen for imprecise excisions from the EP-element line GS3268 from the Kyoto Stock Center using standard techniques[71] and as previously described[1]. The approximate breakpoint locations determined by PCR-analysis are shown in Fig. 2c.

**Antibodies.** Polyclonal antibodies against JIL-1, α-JIL-1 R69, and R70 were described in ref. [6] and Hope in ref. [20]. GST-JASPer (1–475) was used to generate polyclonal antibodies (α-JASPer GP13 and GP14) in guinea pigs (Eurogentech), as

well as the monoclonal (E. Kremmer) antibodies α-JASPer 6F7 and 4D8. α-NDF was a kind gift from J. Kadonaga[72] and GST-MSL3 was used to generate polyclonal antibodies (α-MSL3) in guinea pig (Pineda Antikörper-Service)[73]. The following commercially available antibodies were used: α-H3K36me3 (Abcam, ab9050), α-FLAG (Sigma, F3161), α-H3K9me2 (Abcam, ab1220), α-H3 (Cell Signaling, 9715), α-H3S10ph (Cell Signaling, 9701), α-H4 (Abcam, ab10158), α-H4K16ac (Millipore, 07-329), α-Tubulin (Sigma-Aldrich, T9026), and α-LacI (Millipore, 05-503). For western blots, working concentrations of antibodies were empirically determined (polyclonal sera: 1/500–1/5000; monoclonal culture supernatants: ½–1/10). All antibody dilutions in PBS 3% BSA were reused several times. For detection either the infra-red based Odyssey system (Li-Cor) or the ECL based chemiluminescence system with Chemidoc Touch (Bio-rad) were used.

**Immunofluorescence microscopy of polytene chromosomes.** Immunofluorescence microscopy analysis of polytene chromosome squash preparations was performed as described in ref. [74]. *LacI*-tagged *JIL-1* constructs and the Lac operator insertion line *P11.3* were described in refs. [31,65]. These lines include: *LacI-JIL-1-FL*, *LacI-JIL-1-CTD*, and *LacI-JIL-1-ΔCTD*. GAL4-expression was driven by generating recombinant lines with *Sgs3-GAL4* and *da-GAL4* drivers obtained from the Bloomington Stock Center. Antibody labeling protocols were as in ref. [75]. DNA was visualized by staining with Hoechst 33258 (Molecular Probes) in PBS. The appropriate species-specific and isotype-specific Texas Red-conjugated, TRITC-conjugated, and FITC-conjugated secondary antibodies (Cappel/ICN, Southern Biotech) were used (1:200 dilution) to visualize primary antibody labeling. Mounting of the preparations was in 90% glycerol including 5% n-propyl gallate. Epifluorescence optics were used to examine the preparations on a Zeiss Axioskop microscope. Images were obtained and digitized using a Spot CCD camera. Photoshop (Adobe) was used to pseudocolor, image process, and merge images. Nonlinear adjustments were performed for some images of Hoechst labeling for the best chromosomal visualization.

**JASPer identification.** Nuclear extract from fly embryos were prepared from 12 h embryo collections as described in ref. [76]. For preparative immunoprecipitation (IP), 300 μg nuclear embryo extract 0–12 h at a concentration of 3 mg/mL in HEMG100 buffer (25 mM HEPES pH 7.6, 100 mM KCl, 10% (v/v) glycerol, 0.1 mM EDTA, 12.5 mM MgCl₂) were used per IP. Protein A and Protein G beads mix (1:1) (GE Healthcare) were washed with HEMG100. The diluted extract was pre-cleared with 15 μL (30 μL 50% slurry) Protein A:Protein G beads mix by incubating with end-over-end rotation for 1 h at 4 °C. Beads were pelleted and supernatant was directly used for IP. For IP, the reaction was added to 15 μL (30 μL 50% slurry) Protein A:Protein G beads (GE Healthcare) pre-coupled with antibodies. For pre-coupling, beads were washed with HEMG100 buffer and incubated with end-over-end rotation for 1 h at 4 °C with 2 μg antibodies in HEMG100, using affinity-purified α-JIL-1 R69 and R70 and non-specific rabbit IgG as control. Beads were washed with HEMG100, the extract was added and incubated with end-over-end rotation for 1 h at 4 °C. Beads were spun down and washed three times with HEMG100. Proteins were eluted by incubating beads with HEMG100 supplemented with 0.5% (m/v) N-lauroylsacrosine with end-over-end rotation for 1 h at 4 °C. Proteins were separated by 4–20% gradient SDS-PAGE, stained by Coomassie brilliant blue G250 staining and the most prominent band was cut out for mass spectrometry analysis.

**Immunoprecipitation from embryo extracts.** Nuclear extract from fly embryos were prepared from 12 h embryo collections as described in ref. [76]. For each IP, 400 μg of extract was diluted to 1 mg/mL in BBN buffer (10 mM Tris/Cl pH 8.0, 140 mM NaCl, 1 mM EDTA, 1% (v/v) Triton X-100, 0.1% (v/v) Na deoxycholate, 0.1% (v/v) IGEPAL-CA-360, 0.5 mM DTT) supplemented with cOmplete EDTA-free Protease Inhibitor Cocktail (Sigma-Aldrich). Protein G beads (GE Healthcare) were washed thrice with 10 bed volumes BBN buffer. The diluted extract was pre-cleared with 10 μL (20 μL 50% slurry) Protein G beads by incubating with end-over-end rotation for 1 h at 4 °C. Beads were pelleted at 4 °C for 5 min at 500×g and supernatant was directly used for IP. For IP/MS analysis, IP's were performed from two independent nuclear embryo extracts with two different α-JASPer polyclonal Sera (GP13 and GP14) and two different culture supernatants containing monoclonal antibodies (6F7 and 4D8), as negative control a non-specific serum or culture medium of hybridomas was used. The supernatant was added to 25 μL (50 μL 50% slurry) Protein G beads (GE Healthcare) pre-coupled with antibodies. For pre-coupling, beads were washed thrice with BBN buffer and incubated with end-over-end rotation for 3–4 h at 4 °C with 1.5 mL culture supernatant containing monoclonal antibody and culture medium of hybridomas as control or 2 μL serum in BBN buffer. Beads were washed thrice with BBN buffer, the extract was added and incubated with end-over-end rotation for 3–4 h at 4 °C. Beads were spun down and washed thrice with 40 bed volumes BBN buffer and twice in 10 bed volumes 50 mM Tris/Cl pH 7.5 by incubating with end-over-end rotation for 10 min at 4 °C before handing over to the proteomics core facility.

**Mass spectrometry and data analysis.** Whole IPs were used for trypsin digestion and mass spectrometry (IP/MS) identification of binding partners. For LC-MS/MS purposes, desalted peptides were injected in an Ultimate 3000 RSLCnano system

(Thermo), separated in a 15-cm analytical column (75 μm ID home-packed with ReproSil-Pur C18-AQ 2.4 μm from Dr. Maisch) with a 50-min gradient from 5 to 60% acetonitrile in 0.1% formic acid. The effluent from the HPLC was directly electrosprayed into a Q Exactive HF (Thermo) operated in data-dependent mode to automatically switch between full scan MS and MS/MS acquisition. Survey full scan MS spectra (from $m/z$ 375–1600) were acquired with resolution $R = 60,000$ at $m/z$ 400 (AGC target of $3 \times 10^6$) and MS/MS spectra with resolution 15,000 at $m/z$ 400 (AGC target of $1 \times 10^5$). The 10 most intense peptide ions with charge states between 2 and 5 were sequentially isolated to a target value of $1 \times 10^5$, and fragmented at 27% normalized collision energy. Typical MS conditions were: spray voltage, 1.5 kV; no sheath and auxiliary gas flow; heated capillary temperature, 250 °C; ion selection threshold, 33.000 counts. MaxQuant version 1.5.2.8[77] was used to identify proteins and to quantify by iBAQ with the following parameters: Database, UP000000803_7227_Drome_20160809; MS tol, 10 ppm; MS/MS tol, 20 ppm; Peptide FDR, 0.1; Protein FDR, 0.01 Min. peptide Length, 5; Variable modifications, Oxidation (M); Fixed modifications, Carbamidomethyl (C); Peptides for protein quantitation, razor and unique; Min. peptides, 1; Min. ratio count, 2. The resulting "proteinGroups.txt" file was used for further downstream analysis using DEP version 1.4.0[72] (R) and MSnbase version 2.8.1[78] (R). First, reverse proteins and potential contaminants were removed. The data was filtered for missing values allowing maximally one missing value in at least one condition by calling the function filter_missval (R) (parameter thr = 1). Missing values in control IP samples were considered as missing not at random and imputed using the quantile regression imputation of left-censored data (QRILC) method by calling the function impute (R) (parameter method = "QRILC"). Missing values in the IP samples were considered as missing at random and imputed using the quantile k-nearest neighbor (knn) method by calling the function impute (R) (parameter method = "knn"). To test for statistically significant differentially enriched proteins, the function test_diff (R) was called including condition and sample variables. Proteins were considered as statistically significant enriched with $p$-value < 0.05 and $\log_2$ fold enrichment > 4. GO term analysis of statistical significant enriched proteins was performed using http://www.pantherdb.org using the PANTHER Overrepresentation Test analysis type and PANTHER GO-Slim Biological Process GO terms[79,80]. Protein–Protein interaction network on known interactions of statistically significantly enriched proteins was generated using Cytoscape version 3.7.0[81] and STRING database[82].

**Mapping of protein interactions by co-immunoprecipitation.** Protein-protein interactions were studied with recombinant proteins, in extracts from baculovirus infected Sf21 cells (see also the section of recombinant protein expression). Interaction domains in JASPer and JIL-1 were mapped by co-IP of various truncation mutants using the N-terminal FLAG-tag in JIL-1. JASPer derivatives were all untagged. Interaction of JJ-complex with the core MSL complex (MSL1, MSL2, and MSL3) was analyzed by co-IP from Sf21 cell extracts as described in ref. [46]. In brief, the expression level of all FLAG tagged proteins was assessed by α-FLAG western blot of single extracts and extracts were mixed in order to achieve similar final concentration of the recombinant proteins. IP's were performed with antibodies specific for the specified bait.

**Mononucleosomes, 12-mer nucleosome arrays, and kinase assays.** Mononucleosomes and nucleosome arrays used as substrates for the kinase assays were prepared by salt gradient dialysis as described[34,83]. Briefly, histone octamers (wt and H3K36me3), (biotinylated) scavenger MMTV DNA, and the corresponding 601 DNA[34,84] in 20 mM Tris/HCl, 2 M KCl, 0.1 mM EDTA pH 7.5 at 4 °C were dialyzed into 200 mL nucleosome start buffer (10 mM Tris/HCl, 1.4 M KCl, 1 mM DTT, 0.1 mM EDTA pH 7.5 at 4 °C) for 1 h. 330 mL nucleosome end buffer (10 mM Tris/HCl, 10 mM KCl, 1 mM DTT, 0.1 mM EDTA pH 7.5 at 4 °C) was added overnight at 4 °C using a peristaltic pump (rate 1 mL/min). Subsequently, two additional dialysis steps (4 h and 2 h) were performed using 200 mL nucleosome end buffer. The samples were centrifuged (17,000×$g$, 4 °C, 10 min) and the supernatant isolated. Mononucleosome samples were treated with streptavidin-coated magnetic beads (New England Biolabs) to deplete the biotinylated MMTV DNA and MMTV nucleosomes. All nucleosome arrays were purified by selective MgCl$_2$ precipitation[85].

Recombinant H3 was prepared from inclusion bodies as described in[86]. Prior to the label-free kinase assays, the ratio of JJ-complex to H3 was determined by radioactive kinase assays using γ-ATP and 50 μM non-radioactive ATP in 20 μL total reaction volume as described earlier[6]. Using 2.5 pmol of JJ-complex and 10 pmol of H3 per assay yielded an incorporation of 1 phosphate per H3 molecule. We used the same conditions in the label-free kinase assays with 1 mM of non-radioactive ATP. All reactions were performed in parallel with JJ-complex containing wild type kinase and JJ-complex containing the kinase dead mutant, which is inactive because mutated at both active sites (D407A and D759A). For quantification purposes, we loaded 0.8, 1.6, and 3.2% of the reaction performed with isolated H3, corresponding to 1.2, 2.4, and 4.8 ng, respectively and 30% (3.3 pmol) of the reactions performed with the different types of nucleosomes. The quantitative H3S10ph detection and the loading controls (H4 and H3K36me3) were achieved using IR-coupled secondary antibodies and Odyssey Imaging System (LI-COR).

**Nucleosome pull-down.** Nucleosome library preparation, pull-down experiments and data analysis were performed as described in ref. [34]. Per pull-down reaction, 1.5 pmol of JASPer was used for JASPer-FLAG wt and aromatic cage mutant and for wt and aromatic cage mutant of JJ-complex and pre-coupled to 5 μL FLAG-M2 beads (Sigma-Aldrich) (10 μL 50% slurry) in Binding buffer (20 mM Tris/Cl pH 7.5, 50 mM NaCl, 5 mM EDTA, 0.1% (v/v) TWEEN 20). The protein pre-coupled to beads was incubated with 1.38 pmol nucleosome library containing 115 nucleosome types (12 fmol per nucleosome type) in a total of 200 μL Binding buffer for 4 h at 4 °C with end-over-end rotation. Beads were washed four times with 40 bed volumes (200 μL) Binding buffer and DNA eluted by Proteinase K digestion and purified using a QIAGEN PCR purification kit for further library preparation and sequencing.

**ChIP-seq.** ChIP-seq on MNase-digested chromatin and sonicated chromatin was performed as previously described[46,87]. For spike-in ChIP-seq on MNase-digested chromatin in combination with mild sonication, S2 cells (~$3 \times 10^8$ cells) after RNAi were harvested and cross-linked with 1% formaldehyde for 8 min by adding 1 mL 10× fixing solution (50 mM HEPES pH 8.0, 100 mM NaCl, 1 mM EDTA, 0.5 mM EGTA) with 10% formaldehyde [16% formaldehyde solution (w/v) methanol-fee (Thermo Fischer)] per 10 mL culture at RT. The reaction was stopped by adding 125 mM glycine and incubating for 10 min on ice. Cells were washed twice in PBS and snap-frozen in liquid N$_2$. For nuclei isolation, cells were rapidly thawed and resuspended in PBS supplemented with 0.5% (v/v) Triton X-100 and cOmplete EDTA-free Protease Inhibitor Cocktail (Sigma-Aldrich) (PI) and 5% 79f7Dv3 cells, processed as described for S2 cells without RNAi treatment, relative to S2 cells were added, volume was adjusted to $7 \times 10^7$ cells/mL and cells incubated for 15 min at 4 °C with end-over-end rotation. Nuclei were collected by centrifuging at 4 °C for 10 min at 2000×$g$ and washed once in PBS. For chromatin fragmentation, nuclei were spun down at 4 °C for 10 min at 2000×$g$, resuspended in RIPA (10 mM Tris/HCl pH 8.0, 140 mM NaCl, 1 mM EDTA, 1% (v/v) Triton-X 100, 0.1%(v/v) SDS, 0.1% (v/v) DOC) supplemented with PI and 2 mM CaCl$_2$ at $7 \times 10^7$ cells/mL and digested in 1 mL aliquots by adding 0.6 U MNase (Sigma Aldrich), resuspended in EX50 at 0.6 U/μL[88], and incubated at 37 °C for 35 min with slight agitation. The reaction was stopped by adding 10 mM EGTA and placing on ice. Digested chromatin was sheared with Covaris AFA S220 using 12 × 12 tubes at 50 W peak incident power, 20% duty factor and 200 cycles per burst for 8 min at 5 °C. Subsequent steps were performed as described in ref. [46]. Libraries were prepared with NEBNext Ultra II DNA Library Prep Kit for Illumina (NEB, E7645) and analyzed with 2100 Bioanalyzer with DNA 1000 kit (Agilent). Libraries were sequenced on HiSeq 1500 (Illumina) instrument yielding typically 20–25 million 50 bp single-end reads per sample at the genomics facility.

**RNA-seq.** For RNA-seq, 2 million S2 cells or Kc cells after RNAi treatment were resuspended in Trizol and RNA was purified using the RNeasy Mini Kit (QIAGEN). Afterwards, 1 μg of purified total RNA's was used for rRNA depletion using Ribo-Zero Gold rRNA Removal Kit (Illumnia, MRZG 12324) or NEBNext rRNA Depletion Kit (NEB, E6310). Library preparation was done according to the manufacturer's instructions with NEBNext Ultra II Directional RNA Library Prep Kit for Illumina (NEB, E7760) and analyzed with 2100 Bioanalyzer with DNA 1000 kit (Agilent). Libraries were sequenced on HiSeq 1500 (Illumina) instrument yielding typically 15–50 million 50 bp paired-end reads per sample at the genomics facility.

**NGS data analysis.** Sequencing data were processed using SAMtools version 1.3.1[89], BEDtools version 2.26.0[90], R version 3.5.1 (http://www.r-project.org) and Bioconductor version 3.8 (http://www.bioconductor.org) using default parameters for function calls, unless stated otherwise.

**Read processing.** Sequence reads were aligned to the *D. melanogaster* release 6 reference genome (BDGP6), *D. virilis* FlyBase release r1.07_FB2018_05 reference genome or to *D. melanogaster* transposon sequence set version 9.4.1 (BDGP), including only *D. melanogaster* transposons ($n = 126$), using Bowtie version 1.1.2[91] (parameter –m 1 for *D. melanogaster* genome and transposon) for ChIP-seq and STAR version 2.6.0[92] (parameters --quantMode TranscriptomeSAM GeneCounts, --outFilterMultimapNmax 1) for RNA-seq samples. Gene and transposon quantification of RNA-seq data was performed using RSEM version 1.3.0[93] (parameters –bam, --paired-end, --forward-prob 0).

**RNA-seq analysis.** For RNA-seq analysis, genes and transposons were considered as robustly detected with raw read counts > 0 in all samples. Further analysis was performed using DESeq2 version 1.22.1[94] (R), including blocking variables for batch effect. For transposon analysis, the size factors from the gene analysis were used. Genes and transposons were considered as statistical significant different between conditions with false discovery rate (FDR) < 0.05 by calling the function results (R). For principle component analysis (PCA), a regularized log transformation was applied to per gene counts calling the function rlog (blind = FLASE) (R). To correct for batch effects, the function ComBat (R) was called using a design matrix modeling the RNAi variable by calling the function model.matrix (R). The function plotPCA (R) was called to perform PCA.

**Genome coverage**. ChIP-seq reads were extended to 150 bp and per base normalized genome coverage vectors were calculated as described in ref. [95]. For normalization using *D. virilis* spike-ins, per base coverage vectors were normalized to the sum of *D. virilis* genome coverage vectors multiplied by a factor to adjust for difference in cell number, which was calculated as the total number of reads divided by the total number *D. virilis* reads. To generate non-input normalized per base genome coverage vectors, raw coverage vectors were normalized to million mapped reads (rpm).

**Browser profiles**. Browser profiles were generated by calling the function plot-Profiles from tsTools version 0.1.1 (R) (https://rdrr.io/github/musikutiv/tsTools/) by using mean per base genome coverage vectors after smoothing by computing running medians on 501 bp windows calling the function runmed (endrule = "keep") (R).

**ChIP-seq analysis**. Two H3K36me3 MNase ChIP-seq replicates in Kc and S2 cells each were previously published (GSE94115)[87] and the two Inputs for sonication ChIP-seq (GSE119708)[46]. For gene-centric ChIP-seq analysis, genes were considered as inactive with mean tpm ≤ 1 and active with mean tpm > 1 in control cells. Heat maps of mean normalized coverages at active genes ±500 bp were generated calling the plotRasterHeatmap and convertToColors with using a range of 0.05 to 0.95 from tsTools version 0.1.1 (R) (https://rdrr.io/github/musikutiv/tsTools/). Only genes >3000 bp were considered and the gene body (from TSS +1000 bp to TTS –1000 bp) was scaled to 2000 bp. Exons were converted to per pase gene coverage vectors using a score of 1. Genes were hierarchical clustered on the exon coverages by calculating the Euclidean distance by calling the function dist (R) and clustered using the 'complete' method by calling the function hclust (R). To generate density plots, kernel density estimates were calculated by calling the function density (R). For transposon analysis, the P-element and Penelope transposon were removed as they had zero counts in the input samples, leaving 124 transposons. For calculating ChIP-seq signal at transposons, aligned reads were extended to 150 bp fragments, reads were summed up and normalized with the size factor from reads aligned to the reference genome or for spike-in ChIP-seq to *D. virilis* reference genome. ChIP-seq signal enrichment at transposons was calculated as $log_2$ ratio of IP over input. Heat maps were generated by calling the function pheatmap (R) and smoothed color representation of scatter plots by calling the function smoothScatter (R).

**Reporting summary**. Further information on research design is available in the Nature Research Reporting Summary linked to this article.

## Data availability

The sequencing data discussed in this publication have been deposited in NCBI's Gene Expression Omnibus[96] and are accessible through GEO Series accession number GSE128457 and the mass spectrometry data have been deposited to the ProteomeXchange Consortium[97] with the dataset identifier PXD012790. All other relevant data supporting the key findings of this study are available within the article and its Supplementary Information files or from the corresponding authors upon reasonable request. The source data of cropped images in Figs. 1a, 1b, 1d, 1e, 2c, 2e, and Supplementary Figs. 4e, 4 f, 8d, 13a are provided as Source Data file. A reporting summary for this Article is available as a Supplementary Information file.

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

## Acknowledgements

We thank A. Lukacs, K. Prayitno, and L. Harpprecht for sharing embryo extracts. We thank A. Scacchetti for establishing the *D. virilis* spike-in approach. We thank B.V. Andrianov to kindly share *D. virilis* cell line and J. Kadonaga for NDF reagents. We thank A. Imhof, I. Forné, and M. Wirth at ZFP for mass spectrometry services, S. Krebs and the LAFUGA Genomics Facility for next generation sequencing, T. Straub and the Bioinformatics Unit for providing the high performance computational cluster, E. Kremmer for monoclonal antibody generation, H. Loyd for help with antibody labelings of polytene chromosomes and C. Grimaud for confocal imaging. This work was supported by a grant from the Deutsche Forschungsgemeinschaft to PBB (Be1140/8-1). C.A. acknowledges a DFG fellowship from the Graduate School for Quantitative Biosciences Munich (QBM). Work in the K.M.J. and J.J. laboratory was supported by NIH grant R01 GM62916 and the Roy J. Carver Foundation. National Institutes of Health (NIH) Grants R37 GM086868, R01 GM107047, and P01 CA196539 supported the research in the laboratory of T.W.M. F.W. was funded by a postdoctoral fellowship from the German Research Foundation (WO 2039/1-1).

## Author contributions

C.R. conceived this study and performed experiments. C.A. performed MNase and sonication ChIP-seq experiments and all bioinformatics analysis also with support from T.S. C.W. generated and characterized the cw2 mutant line with help from J.G., W.C. did the LacO-LacI targeting experiments with support from Y.L., and J.J. and K.M.J. supervised the work and secured funding. G.P.D. performed mononucleosome library experiments and F.W. generated the mononucleosomes and arrays for the kinase assays in TWM's lab. S.K. prepared recombinant proteins for all in vitro assays and RNA-seq libraries under the supervision of C.R. and spike-in ChIP-seq experiments under supervision of C.A. S.M. studied RNA binding by JASPer. P.B.B. secured funding and established collaborations. All authors analyzed data. C.R. and C.A. wrote the paper with contributions from all authors.

## Competing interests

The authors declare no competing interests.
