## [Peer Review File · Nature Communications]

Reviewers' comments:

Reviewer #1 (Remarks to the Author):

The work by Albig et al characterises a protein of previously unknown function called JASPer and implies its role in a very interesting phenomenon of gene regulation via H3K36me3, H3S10 phosphorylation, and prevention of heterochromatic spreading. There are interesting implications of this work, and I like the way the overall story is told.

Here are few questions/comments, in no specific order of importance, that need to be addressed before consideration for publication:

1. Throughout the study, the assumption is that JASPer helps JIL-1 targeting to various genomic loci, but nowhere is it shown directly that JIL-1 binding actually changes the H3S10ph status at those loci. A ChIP-seq of H3S10ph, particularly in comparison with the knockdown conditions, would have been a good addition to make this entire study stronger.
2. The authors show down-regulation of TE transcription upon JASPer and JIL-1 depletion, particularly those which have H3K36me3 marks on them. What is the status of H3K36me3 marks in the RNAi scenario.
3. In Fig. 5a, while it is indeed true that the most remarkable change is on chromosome X between male and female cells, the number of genes perturbed in males is more than double compared to female cells even in autosomal regions. What is the possible explanation for this.
4. Line 651: proteins chosen by p-value of <0.05. Do the others mean q-value? Was FDR correction done for this?
5. Line 742: Any specific reason why bowtie version 1 was used when bowtie2 has been released long ago and has been the standard for many years now?
6. Line 777: No reasoning is given for filtering genes <3000 bp.

Reviewer #2 (Remarks to the Author):

In *Drosophila melanogaster*, the essential kinase JIL-1 deposits interphase H3S10ph at actively transcribed genes and is necessary for maintaining chromosomal integrity (Jin et al., 1999). Recruitment of JIL-1 to chromatin remains not well understood – while it is previously known that JIL-1 requires its C' terminal domain (CTD) to bind to chromatin (Bao et al., 2008), the CTD is unlikely to be directly recognizing DNA/nucleosomes as it lacks known chromatin-binding domain. Using immunoprecipitation coupled to mass spectrometry, Albig et al. identified an uncharacterized protein, CG7946/JASPer, that forms a stable complex with JIL-1 and is essential for the stability of JIL-1 and in turn, the deposition of interphase H3S10ph. Interestingly, JASPer contains a PWWP domain and a LEDGF domain which resembles PSIP1/LEDGF in mammals. Albig et al. then mapped the minimal interaction domain to the CTD of JIL-1 and the LEDGF domain of JASPer in vitro. Albig et al. then generated a JASPer-null fly line and showed that JIL-1 and post-mitotic H3S10ph is lost in the salivary glands. Ectopic spreading of H3K9me2 and partial lethality was observed suggesting that JIL-1's function is dependent on JASPer. Interestingly, JASPer also contains a PWWP domain with an aromatic cage recognizing H3K36me3. While in vitro the JIL-1/JASPer kinase did not show detectable preference for H3K36me3-nucleosome arrays, in male S2 cells JIL-1, JASPer, and H3K36me3 co-mark the same genomic regions on autosomes and on chrX. Jil-1 KD causes reduction of JASPer recruitment to the male X, and concomitant increase of H3K9me2. Consistent

with the observation that JIL-1 and JASPer are co-dependent, transcriptome analysis reveals that in both jasper or jil-1 KD, there are more genes down-regulated, especially on the male X chromosome. In addition to its genic effects, JJ complex directly prevents H3K9me2 accumulation at telomeres to maintain expression.

Overall, the study is novel, comprehensive, and addresses the long-standing question of how JIL-1 is recruited to euchromatin region. The biochemical experiments were very thoroughly designed, and I thought the phenocopy of the JASPer mutant to the JIL-1 hypomorph, in particular, was convincing. I have some suggestions to strengthen the major claim of JIL-1's recruitment to H3K36me3 via JASPer's PWWP domain: 1) Recruitment of JIL-1 and H3S10ph was previously demonstrated to be insensitive to heat-shock induced transcription on polytene chromosomes (Cai et al., 2008), given that the JJ complex can apparently bind to nucleosome-free DNA it is important to explicitly address whether JIL-1's recruitment is dependent on H3K36me3, possibly through disruption of set2. 2) In addition, while the authors performed RNA-seq in jasper KD, it would have been more convincing to see ChIP-seq of JIL-1 and H3K9me2 in jasper-depleted cells. Moreover, the authors should consider refining the bioinformatic analysis to delineate direct versus downstream effects of KD, i.e. how many down-regulated genes are occupied by JIL-1? Or gain H3K9me2? Similar to what was done for transposable elements, there are clearly subset of genes that are more sensitive to loss of JJ-complex – can they be explained by JJ complex enrichment? Or proximity to a H3K9me2-marked domain or transposon?

Minor comments

- D. virilis normalization of ChIP-seq – normalization ratio should be explicitly calculated from input stores instead of assuming 5% for all chromatin preps.
- Consider adding a panel in Fig. 2 for the scoring of sub-lethality phenotype of jaspercw2/cw2
- Fig. 2F – the RNA-seq changes for jil1 and jasper in the knockdown experiment does not seem to be significant. Was statistical testing performed? Why is there a discrepancy between protein level (Fig. 1E) and RNA expression?
- Fig. 3A/B – PWWP domains have been demonstrated to recognize H3K36me2/3. Was binding to H3K36me2 nucleosomes tested?
- Fig. 5. How many differentially expressed genes are shared (i.e. shows the same direction of change) in jil-1 and jasper KD? If there are any, what is the function of those genes?
- Fig. 7A. It's misleading to label the JASPER IP/MS results as JJ-complex interactome, as these interactions may not be dependent on the presence of JIL-1.
- Fig. S11. Accumulation of H3K9me2 at telomeric HTT arrays in the jil-1 KD is not convincing and authors should consider changing the figure title.
- Fig. S12. H3S10ph does not seem to change in the jasper KD S2 cells, despite the dramatic depletion of JIL-1 and JASPER by western blot in Fig. 2E. Was this consistently observed? If so, this should be explicitly stated in the text to interpret the S2 knockdown results.

Bao, X., Cai, W., Deng, H., Zhang, W., Krencik, R., Girton, J., et al. (2008). The COOH-terminal domain of the JIL-1 histone H3S10 kinase interacts with histone H3 and is required for correct targeting to chromatin. *Journal of Biological Chemistry*, 283(47), 32741–32750.

<http://doi.org/10.1074/jbc.M806227200>

Cai, W., Bao, X., Deng, H., Jin, Y., Girton, J., Johansen, J., & Johansen, K. M. (2008). RNA polymerase II-mediated transcription at active loci does not require histone H3S10 phosphorylation in *Drosophila*. *Development*, 135(17), 2917–2925.

<http://doi.org/10.1242/dev.024927>

Jin, Y., Wang, Y., Walker, D. L., Dong, H., Conley, C., Johansen, J., & Johansen, K. M. (1999). JIL-1: a novel chromosomal tandem kinase implicated in transcriptional regulation in *Drosophila*. *Molecular Cell*, 4(1), 129–135.

Reviewer #3 (Remarks to the Author):

Review of the manuscript entitled JASPer controls interphase histone H3S10 phosphorylation by chromosomal kinase JIL-1 in *Drosophila*, authored by Christian Albig and collaborators. The manuscript is well written, the hypothesis are well explained and the rationale behind is clear. The experiments are well justified and the results support the taken conclusions. In my opinion this manuscript should be accepted for publication in Nature Communications after minor revisions which, I think would enrich the discussion opened by the manuscript. Specific comments as follows:

1) Introduction; Lines 97-98: " We propose that JIL-1 regulates interphase chromatin structure and function through H3S10 phosphorylation in collaboration with other enzymes". This statement at the end of the introduction is not a novelty, since it has been proposed before and the authors acknowledge it at the beginning of the introduction as the current hypothesis for the function of Jil-1. Therefore, I suggest that they modify this statement by including the role of Jasper together with Jil-1 which, is the actual novelty that they are reporting.

2) Results

- Inconsistency in the size of Jasper between the western blot of Figure 1a) and the comassie gel of the same figure section b)
- Redundancy in Suppl. Figure 1b and Figure 1c
- Figure 2 e and f, results show different outcomes of RNAi at mRNA and protein levels. In e) RNAi of Jasper abrogates both Jasper and Jil-1 protein in both S2 and Kc cells while in f) Jasper RNAi shows Jil-1 mRNA in Kc cells. The authors explain this in the text by discarding transcription regulation. Nevertheless, Jil-1 RNAi does not affect protein levels of Jasper in either Kc and S2 cells (e) it seems that Jil-1 RNAi clearly affects the levels of Jasper mRNA in both cells (f). The authors don't comment on this.
- Related to the above. The Cw2 mutant of Jasper eliminates Jil-1, but shows only partial lethality. On the other hand the z2 null allele of Jil-1 shows stronger lethality? how come in a scenario of the cw2 where Jil-1 is undetectable, isn't lethal? Authors should discuss on this
- For the section entitled The JJ-complex localizes to active chromatin in vivo and its enrichment on the male X chromosome depends on JIL-1 and dosage compensation Could the authors also comment of the MSL quantification and distribution in Jil-1 mutants? (Polytenes). Have they look into it in addition to the experiments that they present here. It would be interesting and it is not difficult to implement and would reinforce their hypothesis of the JJ-complex and the MSL-complex
- Western blot from co-ip of MSL and JIL-1 in suppl.figure 8d: no molecular weight marker is shown. Bands corresponding to Jasper in the case of MSL IP are not clearly shown
- It would be interesting to know where do the other retrotransposons that show a considerable enrichment on H3 K36 me3, in addition to the elements in the H-T-T array, the GYPSY 5 and the 3S18, are located. Are these regions with euchromatin/heterochromatin boundaries?
- I do see an inconsistency between the increased amounts of H3K9me2 in Jil-1 RNAi shown in Figure 6 c, and the amounts shown by the chip of the Jil-1 RNAi in Suppl.figure 11. Can the values shown in Figure 6 c come from the heat map shown in Suppl.Figure 11? If it is difficult to see the enrichment on this mark, maybe the authors could chip other heterochromatin proteins as HP1 or chromatin marks like H3K9me3?

Elena Casacuberta, PhD
CSIC Scientist

Institut of Evolutionary Biology, IBE
CSIC-Pompeu Fabra University (UPF)
Passeig Marítim de la Barcelona, 37-49.
E-08003 Barcelona (Spain)
Tel: +34 932309637
<http://biologiaevolutiva.org/ecasacuberta/>
elena.casacuberta@ibe.upf-csic.es

Answer to the reviewers

Reviewer #1 (Remarks to the Author):

The work by Albig et al characterises a protein of previously unknown function called JASPer and implies its role in a very interesting phenomenon of gene regulation via H3K36me3, H3S10 phosphorylation, and prevention of heterochromatic spreading. There are interesting implications of this work, and I like the way the overall story is told.

Response: We thank the reviewer for the appreciation and the careful assessment of the paper.

Here are few questions/comments, in no specific order of importance, that need to be addressed before consideration for publication:

1. Throughout the study, the assumption is that JASPer helps JIL-1 targeting to various genomic loci, but nowhere is it shown directly that JIL-1 binding actually changes the H3S10ph status at those loci. A ChIP-seq of H3S10ph, particularly in comparison with the knockdown conditions, would have been a good addition to make this entire study stronger.

Response: The reviewer suggested an important experiment that we have performed, but unfortunately did not reveal a reduction of H3S10ph at JJ-complex

binding sites when the complex is depleted. As we have argued earlier (Regnard et al., 2011), we think this is most likely explained by the fact that H3S10ph levels are dominated by mitotic H3S10ph in exponentially growing cells, which is at least two orders of magnitude higher than interphase H3S10ph. JJ-complex depletion does not affect these levels, which are uniformly distributed along all chromosomes. Unfortunately, *Drosophila* cells cannot be sorted to sufficient homogeneity with respect to cell cycle stage.

To overcome this technical problem, we resorted to polytene chromosomes, where interphase chromatin can be studied without mitotic contributions. In Fig. 2 we showed that H3S10ph is absent in the *JASPer*^{cw2/cw2} mutant. In our opinion, this fact, albeit at low resolution, combined with our (Johansen's Lab) earlier characterization of the tight link between interphase H3S10ph, JIL-1 and H3K9me2 spreading in flies (Cai et al., 2014; Wang et al., 2012) makes a very strong case for our conclusions.

We amended the text (L149 and L290) to alert the readers to these facts.

See also the answer to minor point on Supplementary Figure 12 by reviewer #2.

2. The authors show down-regulation of TE transcription upon JASPer and JIL-1 depletion, particularly those which have H3K36me3 marks on them. What is the status of H3K36me3 marks in the RNAi scenario.

Response: It is well established that H3K36me3 is placed co-transcriptionally by the set2 histone methyl transferase. We have shown that the autosomal genic enrichment of JASPer is unchanged upon JIL-1 KD, which strongly suggesting that H3K36me3 is unchanged genome-wide.

The picture is a bit different for telomeric transposons. JASPer is still enriched at the transposons of the HTT arrays after *jil-1* RNAi, but the levels are reduced. This strongly suggests that H3K36me3 is still present but might be reduced as a consequence of the decreased transcription.

We clarified the text accordingly in L327: "...suggesting that either H3K36me3 is decreased there because of the lower expression and/or JIL-1 contributes to the enrichment of the JJ-complex at telomeres."

3. In Fig. 5a, while it is indeed true that the most remarkable change is on chromosome X between male and female cells, the number of genes perturbed in males is more than double compared to female cells even in autosomal regions. What is the possible explanation for this?

Response: We think that the observed statistical difference in transcription changes of autosomal genes between males and females lacks biological significance. The total number of genes called to be statistical significantly different between conditions not only depends on the difference (\log_2 fold change), but also on the variance (which is the dispersion parameter in the

DESeq2 package). In the Kc cell data set, the dispersion is double compared to S2 cells, which implies that fewer genes are called significant. Therefore, we believe that the difference between S2 and Kc cells is not biologically meaningful.

4. Line 651: proteins chosen by p-value of <0.05 . Do the others mean q-value? Was FDR correction done for this?

Response: We mean p-values, which were not adjusted for multiple comparison. We decided to use non-adjusted p-values in addition to a \log_2 fold change cutoff, as we use mass spectrometry analysis as an exploratory tool to identify potential new interactors. Based on the mass spectrometry data, we can formulate new hypotheses which must be validated in the future. In addition, we applied a \log_2 fold change cutoff (which controls for effect size) to make sure that only the top enriched proteins are considered. Other proteins might still be significant but less enriched, therefore we would consider those less relevant.,

5. Line 742: Any specific reason why bowtie version 1 was used when bowtie2 has been released long ago and has been the standard for many years now?

Response: There are two main reasons why we used bowtie1. First, bowtie1 can be faster and/or more sensitive for reads up to 50 bp (here, we used 50 bp single reads for ChIP-seq) relative to bowtie2. Second, we wished to remove multi-mapping reads by using bowtie1's $-m$ parameter, which is not available for bowtie2.

6. Line 777: No reasoning is given for filtering genes <3000 bp.

Response: For Supplementary Figure 6d, we focused on genes >3000 bp as we intended to show that H3K36me3, JASPer and JIL-1 are enriched on the exons of expressed genes. To best visualize a different ChIP-seq signal between exons and introns at the resolution of our MNase ChIP-seq experiment, genes should harbor longer introns, which are more abundant in longer genes. Furthermore, we used a heatmap representation with unscaled regions around TSS + 1000 bp and TTS - 1000 bp, which required that the gene is longer as 2000 bp. In addition, we scaled the intervening "gene body" to 2000 bp and, to exclude that we observe strange signals due to rescaling, we chose that the "gene body" should also contain at least 1000 bp.

In the legend of Supplementary Figure 6d we changed the text to:

"Genes > 3000 bp (n=3152), which often harbor long introns at their 5' ends, were considered to better visualize intron-exon boundaries."

Reviewer #2 (Remarks to the Author):

In *Drosophila melanogaster*, the essential kinase JIL-1 deposits interphase

H3S10ph at actively transcribed genes and is necessary for maintaining chromosomal integrity (Jin et al., 1999). Recruitment of JIL-1 to chromatin remains not well understood – while it is previously known that JIL-1 requires its C' terminal domain (CTD) to bind to chromatin (Bao et al., 2008), the CTD is unlikely to be directly recognizing DNA/nucleosomes as it lacks known chromatin-binding domain. Using immunoprecipitation coupled to mass spectrometry, Albig et al. identified an uncharacterized protein, CG7946/JASPer, that forms a stable complex with JIL-1 and is essential for the stability of JIL-1 and in turn, the deposition of interphase H3S10ph. Interestingly, JASPer contains a PWWP domain and a LEDGF domain which resembles PSIP1/LEDGF in mammals. Albig et al. then mapped the minimal interaction domain to the CTD of JIL-1 and the LEDGF domain of JASPer in vitro. Albig et al. then generated a JASPer-null fly line and showed that JIL-1 and post-mitotic H3S10ph is lost in the salivary glands. Ectopic spreading of H3K9me2 and partial lethality was observed suggesting that JIL-1's function is dependent on JASPer. Interestingly, JASPer also contains a PWWP domain with an aromatic cage recognizing H3K36me3. While in vitro the JIL-1/JASPer kinase did not show detectable preference for H3K36me3-nucleosome arrays, in male S2 cells JIL-1, JASPer, and H3K36me3 co-mark the same genomic regions on autosomes and on chrX. Jil-1 KD causes reduction of JASPer recruitment to the male X, and concomitant increase of H3K9me2. Consistent with the observation that JIL-1 and JASPer are co-dependent, transcriptome analysis reveals that in both jasper or jil-1 KD, there are more genes down-regulated, especially on the male X chromosome. In addition to its genic effects, JJ complex directly prevents H3K9me2 accumulation at telomeres to maintain expression.

Overall, the study is novel, comprehensive, and addresses the long-standing question of how JIL-1 is recruited to euchromatin region. The biochemical experiments were very thoroughly designed, and I thought the phenocopy of the JASPer mutant to the JIL-1 hypomorph, in particular, was convincing. I have some suggestions to strengthen the major claim of JIL-1's recruitment to H3K36me3 via JASPer's PWWP domain:

Response: We thank the reviewer for the laudatory remarks and we appreciate the suggestions for improvement.

1) Recruitment of JIL-1 and H3S10ph was previously demonstrated to be insensitive to heat-shock induced transcription on polytene chromosomes (Cai et al., 2008), given that the JJ-complex can apparently bind to nucleosome-free DNA it is important to explicitly address whether JIL-1's recruitment is dependent on H3K36me3, possibly through disruption of set2.

Response: We think that in the context of what is known about PWWP domains our data are very consistent with a major contribution of H3K36me3 to JASPer

recruitment. The observed non-specific DNA binding is in line with this recruitment mode, since structural analysis and modeling of the closely related PWWP domain of psip1/LEDGF reveals that H3K36me3 recognition involves non-specific contact with DNA on the nucleosome through a highly positive charged surface (see Figure 6 in (Eidahl et al., 2013)). We mention this in L197-203 of the manuscript.

Concerning the heat shock experiments, we previously showed that in cells JIL-1 is present at the HSP70 gene before heat shock and is decreased upon heat shock (Fig. S10 in (Regnard et al., 2011)). This observation is again in line with our current conclusions: H3K36me3 is associated with the known leaky transcription of the hsp70 gene prior to induction, which recruits JIL-1. The loss of JIL-1 correlates with the massive loss of nucleosomes observed upon heat shock (Petsch et al., 2008).

Some time ago we depleted *set2* by RNAi in S2 cells. However, the maximal depletion of H3K36me3 we achieved was only of 50-60%. ChIP-qPCR analysis showed that JASPer and JIL-1 enrichment at gene bodies was decreased (see reviewer Fig. 1, at end of this rebuttal). While these data are clearly supportive of our current model, we do not wish to add them to the manuscript now, since they were generated with older methodology that we do not consider state of art anymore. We are also planning a more extensive dissection of the recruitment modes of the JJ-complex using (combined) RNAi of various H3K36 methyltransferases and/or interactors in addition to RNase treatment, to explain in particular the two-fold enrichment of JIL-1 on the X chromosome, which we cannot explain now. We think that this work is beyond the scope of this manuscript.

We do not wish to exclude that additional principles contribute to JJ-complex enrichment at particular sites, therefore we modified the text in L397-400 accordingly:

“...we do not exclude that additional binding modes are relevant at promoters and enhancers as described earlier (Cai et al., 2014; Kellner, Ramos, Van Bortle, Takenaka, & Corces, 2012). Those binding modes could implicate interactions with other chromatin complexes, RNA or DNA.”

2) In addition, while the authors performed RNA-seq in jasper KD, it would have been more convincing to see ChIP-seq of JIL-1 and H3K9me2 in jasper-depleted cells.

Response: Because upon depletion of JASPer, JIL-1 protein is unstable and cannot be detected on Western blots we do not think the ChIP experiment would be informative as the ChIP bait would be missing.

3) Moreover, the authors should consider refining the bioinformatic analysis to

delineate direct versus downstream effects of KD, i.e. how many down-regulated genes are occupied by JIL-1? Or gain H3K9me2? Similar to what was done for transposable elements, there are clearly subset of genes that are more sensitive to loss of JJ-complex – can they be explained by JJ complex enrichment? Or proximity to a H3K9me2-marked domain or transposon?

Response: We initially did not correlate the changes in steady state RNA levels and the chromatin changes upon JIL-1 depletion, mostly because in both cases the changes are very small. We now explored the correlation of RNA-seq expression changes to changes in H3K9me2/H4K16ac/JASPer ChIP-seq after *jil-1* RNAi.

We found that X-linked genes losing H4K16ac are mostly down-regulated and that X-linked genes gaining H3K9me2 are also mostly down-regulated. Similarly, genes on the X chromosome losing JASPer binding are also mostly down-regulated. Expectedly, there is no linear correlation since the two histone marks and JASPer (and H3K36me3) are not directly proportional (or inverse-proportional in the case of H3K9me2) to the steady state RNA levels. We have shown this in particular for H3K36me3 previously (Fig. S6 in (Regnard et al., 2011)). We added the corresponding scatter/density plots to a new Supplementary Figure 10 and changed the text in the results L296-298 as follows: “In a gene-by-gene analysis of X-linked genes, we observed that down-regulated genes consistently tend to loose H4K16ac and JASPer or gain H3K9me2 (Supplementary Fig.S10).

However, we did not find a direct relationship between the expression changes and the distance to heterochromatin. There was also no correlation of the expression changes to the distance to high affinity sites (HAS). This is in accordance with the small change of H4K16ac density at active genes which also does not correlate to distance from HAS upon *jil-1* RNAi (Supplementary Fig. 8). Furthermore, we cannot correlate expression changes to the density of JJ-complex at active genes in the control. Nearly all robustly detected transcripts, which were used for our RNA-seq analysis, are encoded by genes enriched for the JJ-complex and the density of JASPer and JIL-1 is similar at responding and non-responding genes. Of course, we can provide all the plots on request of the reviewer.

Minor comments

- D. virilis normalization of ChIP-seq – normalization ratio should be explicitly calculated from input stores instead of assuming 5% for all chromatin preps.

Response: The reviewer is right that we could use the actual ratio instead of assuming 5%. We implemented this step now in our analysis. However, we do not see changes in ChIP-seq coverage due to the applied arcsine square root transformation and standardization.

- Consider adding a panel in Fig. 2 for the scoring of sub-lethality phenotype of *japsercw2/cw2*

Response: We added a corresponding table in Figure 2e.

- Fig. 2F – the RNA-seq changes for *jil1* and *japser* in the knockdown experiment does not seem to be significant. Was statistical testing performed? Why is there a discrepancy between protein level (Fig. 1E) and RNA expression?

Response: There is a misunderstanding because we do not show the KD efficiency of *JIL-1* and *JASPer* at the RNA level. We cannot properly show the depletion at the mRNA level because we perform rRNA depletion for our RNA-seq (not polyA –selected) and thus also sequence plenty of the dsRNA used for the KD. We also performed a statistical test on the fold change of expression values and observed no significant deviation from zero (p-value > 0.05). We included the p-values in the figure. The purpose of Figure 2f is to illustrate that we do not see any reduction on *JIL-1* mRNA abundance in *jasper* RNAi, while we observe a loss of JIL-1 protein in *jasper* RNAi. This argues that JIL-1 protein is unstable in the absence of its binding partner JASPer.

-Fig. 3A/B – PWWP domains have been demonstrated to recognize H3K36me2/3. Was binding to H3K36me2 nucleosomes tested?

Response: We did not test binding of JASPer to mononucleosomes bearing H3K36me2 mark because it was not included in the original library already containing 115 different barcoded DNA/nucleosomes. There is no H3K36me2 ChIP-seq profile from S2 cells available in modENCODE that we could correlate to JJ-complex.

We still think it is likely that JASPer prefers H3K36me3 over H3K36me2:

Reviewer Fig. 1 suggests that upon Set2 KD, there is a tendency for H3K36me2 to be increased at the tested loci, as expected if the conversion of me2 to me3 is inhibited, but JASPer ChIP is clearly decreased and thus correlates best with me3.

-Fig. 5. How many differentially expressed genes are shared (i.e. shows the same direction of change) in *jil-1* and *jasper* KD? If there are any, what is the function of those genes?

Response: In Supplementary Figure 9b we show that 592 transcripts are down-regulated in both RNAi depletions of S2 cells, whereas 946 and 977 genes are down-regulated in *jil-1* and *jasper* KD, respectively (Fig. 5a). Overall, the expression changes between *jil-1* and *jasper* depletions correlate well ($r = 0.597$). The same is true for Kc cells. One third of the down-regulated genes in S2 cells are on the X-chromosome (Figure 5a) leading to an overall reduction of X-linked transcripts (Figure 5b).

A GO analysis on common down-regulated genes in S2 cells showed that the most significant GO terms are just below FDR < 0.05 and rather broad terms (i.e.

cellular process, localization, and vesicle mediated transport), which are inconclusive.

- Fig. 7A. It's misleading to label the JASPER IP/MS results as JJ-complex interactome, as these interactions may not be dependent on the presence of JIL-1.

Response: We modified the title of Figure 7 in L1254 to “The JASPer interaction network and other H3K36me3 binding complexes in *Drosophila melanogaster*”

Fig. S11. Accumulation of H3K9me2 at telomeric HTT arrays in the *jil-1* KD is not convincing and authors should consider changing the figure title.

Response: See also answer to reviewer #3 last point concerning Supplementary Fig. 11. We replaced the heatmap with the log₂ fold changes.

The title of new Supplementary Fig. 12 has been changed to: “Change in JASPer and H4K16ac enrichment at the telomeric transposons upon JIL-1 depletion in S2 cells.”

- Fig. S12. H3S10ph does not seem to change in the *jasper* KD S2 cells, despite the dramatic depletion of JIL-1 and JASPER by western blot in Fig. 2E. Was this consistently observed? If so, this should be explicitly stated in the text to interpret the S2 knockdown results.

Response: As we have argued earlier this is most likely explained by the fact that H3S10ph levels are dominated by mitotic H3S10ph in exponentially growing cells, which is at least two orders of magnitude higher than interphase H3S10ph. JJ-complex depletion does not affect mitotic H3Ser10ph, which is uniformly distributed along all chromosomes. Unfortunately, *Drosophila* cells cannot be sorted to sufficient homogeneity with respect to cell cycle stage.

See also the answer to point 1) by reviewer#1.

Bao, X., Cai, W., Deng, H., Zhang, W., Krencik, R., Girton, J., et al. (2008). The COOH-terminal domain of the JIL-1 histone H3S10 kinase interacts with histone H3 and is required for correct targeting to chromatin. *Journal of Biological Chemistry*, 283(47), 32741–32750. <http://doi.org/10.1074/jbc.M806227200>

Cai, W., Bao, X., Deng, H., Jin, Y., Girton, J., Johansen, J., & Johansen, K. M. (2008). RNA polymerase II-mediated transcription at active loci does not require histone H3S10 phosphorylation in *Drosophila*. *Development*, 135(17), 2917–2925. <http://doi.org/10.1242/dev.024927>

Jin, Y., Wang, Y., Walker, D. L., Dong, H., Conley, C., Johansen, J., & Johansen, K. M. (1999). JIL-1: a novel chromosomal tandem kinase implicated in transcriptional regulation in *Drosophila*. *Molecular Cell*, 4(1), 129–135.

Reviewer #3 (Remarks to the Author):

Review of the manuscript entitled JASPer controls interphase histone H3S10 phosphorylation by chromosomal kinase JIL-1 in *Drosophila*, authored by Christian Albig and collaborators.

The manuscript is well written, the hypothesis are well explained and the rationale behind is clear. The experiments are well justified and the results support the taken conclusions. In my opinion this manuscript should be accepted for publication in Nature Communications after minor revisions which, I think would enrich the discussion opened by the manuscript. Specific comments as follows:

Response: We thank the reviewer for the positive suggestions and comments that helped to improve the paper further.

1) Introduction; Lines 97-98: “ We propose that JIL-1 regulates interphase chromatin structure and function through H3S10 phosphorylation in collaboration with other enzymes”. This statement at the end of the introduction is not a novelty, since it has been proposed before and the authors acknowledge it at the beginning of the introduction as the current hypothesis for the function of Jil-1. Therefore, I suggest that they modify this statement by including the role of Jasper together with Jil-1 which, is the actual novelty that they are reporting.

Response: We thank the reviewer for its scrutiny and deleted the last sentence of the introduction L95.

2) Results

- Inconsistency in the size of Jasper between the western blot of Figure 1a) and the comassie gel of the same figure section b)

Response: We changed the picture of the recombinant JJ-complex in Fig. 1b and corrected the mislabeling and corresponding legend. JASPer is actually the lower band, and the other band is a contaminant we have in many purifications. MWM are actually right.

- Redundancy in Suppl. Figure 1b and Figure 1c

Response: The reviewer is right, we do not mention the *Su(var)3-1* alleles of JIL-1 anywhere else than in the introduction. We eliminate Supplementary Fig. 1b.

- Figure 2 e and f, results show different outcomes of RNAi at mRNA and protein levels. In e) RNAi of Jasper abrogates both Jasper and Jil-1 protein in both S2 and Kc cells while in f) Jasper RNAi shows Jil-1 mRNA in Kc cells. The authors explain this in the text by discarding transcription regulation. Nevertheless, Jil-1 RNAi does not affect protein levels of Jasper in either Kc and S2 cells (e) it seems that Jil-1 RNAi clearly affects the levels of Jasper mRNA in both cells (f). The authors don't comment on this.

Same answer as to the minor point to Fig.2f from reviewer#2:

Response: There is a misunderstanding because we do not show the KD efficiency of JIL-1 and JASPer at the RNA level. We cannot properly show the depletion at the mRNA level because we do rRNA depletion for our RNA-seq (not polyA –selected) and thus also sequence plenty of the dsRNA used for the KD. We also performed a statistical test on the fold change of expression values and observed no significant deviation from zero (p-value > 0.05). We included the p-values in the figure. The purpose of Figure 2f is to illustrate that we do not see any reduction on *JIL-1* mRNA abundance in *jasper* RNAi, while we observe a loss of JIL-1 protein in *jasper* RNAi. This argues that JIL-1 protein is unstable in the absence of its binding partner JASPer.

- Related to the above. The Cw2 mutant of Jasper eliminates Jil-1, but shows only partial lethality. On the other hand the z2 null allele of Jil-1 shows stronger lethality? how come in a scenario of the cw2 where Jil-1 is undetectable, isn't lethal? Authors should discuss on this

Response: We believe that there is residual JIL-1 kinase activity in the *JASPer^{cw2/cw2}* mutant, which may still fulfill some functions. In the absence of JASPer, JIL-1 is still produced but appears unstable. However, trace amounts of active kinase or fragments of it might be sufficient to allow a better viability. It has been described in Li et al, JBC, 2013 for the rescue of the *JIL-1^{z2/z2}* mutant:

TABLE 1
Properties of JIL-1 constructs expressed in a *JIL-1^{z2}/JIL-1^{z2}* null background

Construct	Localization to chromatin	Rescue of autosome morphology	H3S10 phosphorylation	H3K9me2 spreading	Rescue of adult viability* (n)
JIL-1-FL	Yes	Yes	Yes	No	58.6 (644)
CTD	Yes	Yes	No	Yes	13.6 (515)
ΔCTD	Yes ^b	Yes	Yes	No	4.4 (591)
ΔNTD	Yes	Yes	No	Yes	18.9 (428)
KDI*/KDII*	Yes	No	No	Yes	22.9 (447)
KDI*/KDII	Yes	No	No	Yes	14.3 (329)
KDI/KDII*	Yes	No	No	Yes	12.3 (568)
JIL-1 ^{S424A}	Yes	Partial	No	Yes	14.7 (439)

* The rescue of adult viability in the *JIL-1* null mutant background by each construct was calculated as described under "Experimental Procedures." n indicates the total number of eclosed progeny in each experiment.

^b Ectopic chromatin localization.

We clarified this issue in the text L160 by adding the sentence:

" However, trace amounts of JIL-1 or fragments of it might still be expressed and could explain the better viability of *JASPer^{cw2/cw2}* mutant versus *JIL-1^{z2/z2}* mutant (Li et al., 2013)"

- For the section entitled The JJ-complex localizes to active chromatin in vivo and its enrichment on the male X chromosome depends on JIL-1 and dosage compensation Could the authors also comment of the MSL quantification and distribution in Jil-1 mutants? (Polytenes). Have they looked into it in addition to the experiments that they present here. It would be interesting and it is not difficult to implement and would reinforce their hypothesis of the JJ-complex and the MSL-complex

Response: The reviewer raises an important point which we did not mention in the manuscript because of space limitation. Quantification on polytene

chromosomes is difficult in particular in the *JIL-1* null background since the banded pattern of the chromosomes is lost. However, MSL2 still binds the X chromosome in hypomorph and *JIL-1* null mutant backgrounds (Wang, 2001, Deng, 2005). In the *JASPer*^{cw2/cw2} it appears that the dosage compensation complex spreading on the male X chromosome is normal. We chose to present only the distribution at high resolution obtained by ChIP-seq focusing on MSL3 as a proxy for the localization of the dosage compensation complex and on H4K16ac as a proxy for dosage compensation efficiency. There are only slight changes in the absence of JIL-1, which confirms the genetics, namely that the distribution of the dosage compensation complex is mostly independent of JIL-1, whereas JIL-1 needs the complex to be enriched on the X chromosome. See also the answer to point 3. from reviewer #2 asking for more bioinformatics analysis and the new Supplementary Fig. 10.

- Western blot from co-ip of MSL and JIL-1 in supplementary figure 8d: no molecular weight marker is shown. Bands corresponding to Jasper in the case of MSL IP are not clearly shown

Response: To clarify Supplementary Fig.8d, we put the two panels on the top of each other to magnify them and added the marker on the lower panel. As illustrated in the anti-Flag western blot on the top, FLAG-JIL-1 and FLAG-MSL1 run together, and FLAG-MSL3 and untagged JASPer are very close. FLAG-MSL3 runs only slightly slower than JASPer and is visible in the window used for JASPer, even when the gels run longer to increase the separation.

- It would be interesting to know where do the other retrotransposons that show a considerable enrichment on H3K36me3, in addition to the elements in the H-T-T array, the GYPSY 5 and the 3S18, are located. Are these regions with euchromatin/heterochromatin boundaries?

Response: gypsy 5 and 3S18 are now labeled in the plots of Fig. 6b and Supplementary Fig. 11. Both are LTR-transposons and according to the counts we detect in our input genomic DNA for all TE's they are among the abundant ones (3S18 > 10000 rpkm and gypsy5 >1000 rpkm), suggesting that the enrichment of H3K36me3 and JJ-complex is probably real. However, while we can estimate the copy number of each element using our input genomic DNA, we do not know where each single copy is located in the genome.

- I do see an inconsistency between the increased amounts of H3K9me2 in Jil-1 RNAi shown in Figure 6 c, and the amounts shown by the chip of the Jil-1 RNAi in Suppl.figure 11. Can the values shown in Figure 6 c come from the heat map shown in Suppl.Figure 11? If it is difficult to see the enrichment on this mark, maybe the authors could chip other heterochromatin proteins as HP1 or chromatin marks like H3K9me3?

Response: In Fig. 6c we show the \log_2 fold change of H3K9me2 signal between RNAi and control condition, while in old Supplementary Fig.11 we show the absolute \log_2 enrichment of H3K9me2 in RNAi and control condition separately. The coloring of the heatmap in old Supplementary Fig. 11 is dominated by broad dynamic range of \log_2 enrichment values between individual TEs, roughly between the range of -2 and +3. Therefore, the difference of \log_2 fold change of < 0.5 between RNAi and control condition, which we observe for TEs of the HTT arrays, is challenging to visualize in this representation. To avoid confusion, we replaced the heatmap in new Supplementary Fig. 12 by two bar charts showing JASPer and H4K16ac \log_2 fold changes between *jil-1* RNAi and control.

- Cai, W., Wang, C., Li, Y., Yao, C., Shen, L., Liu, S., . . . Johansen, K. M. (2014). Genome-wide analysis of regulation of gene expression and H3K9me2 distribution by JIL-1 kinase mediated histone H3S10 phosphorylation in Drosophila. *Nucleic Acids Res*, *42*(9), 5456-5467. doi:10.1093/nar/gku173
- Eidahl, J. O., Crowe, B. L., North, J. A., McKee, C. J., Shkriabai, N., Feng, L., . . . Kvaratskhelia, M. (2013). Structural basis for high-affinity binding of LEDGF PWWP to mononucleosomes. *Nucleic Acids Res*, *41*(6), 3924-3936. doi:10.1093/nar/gkt074
- Kellner, W. A., Ramos, E., Van Bortle, K., Takenaka, N., & Corces, V. G. (2012). Genome-wide phosphoacetylation of histone H3 at Drosophila enhancers and promoters. *Genome Res*, *22*(6), 1081-1088. doi:10.1101/gr.136929.111
- Li, Y., Cai, W., Wang, C., Yao, C., Bao, X., Deng, H., . . . Johansen, K. M. (2013). Domain requirements of the JIL-1 tandem kinase for histone H3 serine 10 phosphorylation and chromatin remodeling in vivo. *J Biol Chem*, *288*(27), 19441-19449. doi:10.1074/jbc.M113.464271
- Regnard, C., Straub, T., Mitterweger, A., Dahlsveen, I. K., Fabian, V., & Becker, P. B. (2011). Global analysis of the relationship between JIL-1 kinase and transcription. *PLoS Genet*, *7*(3), e1001327. doi:10.1371/journal.pgen.1001327
- Wang, C., Cai, W., Li, Y., Girton, J., Johansen, J., & Johansen, K. M. (2012). H3S10 phosphorylation by the JIL-1 kinase regulates H3K9 dimethylation and gene expression at the white locus in Drosophila. *Fly (Austin)*, *6*(2), 93-97. doi:10.4161/fly.20029

Reviewer Figure 1. ChIP of H3K36me3, H3K36me2, JIL-1 and JASPer after Set2 RNAi

S2 cells were treated for 7 days with 2 different dsRNA (#1 and #2) for Set2 knock down and a control dsRNA (GST). ChIP were performed as in Regnard, et al, 2011, and analyzed by q-PCR as % of input. Westernblot analysis showed a 50-60% reduction of bulk H3K36me3 (not shown).

REVIEWERS' COMMENTS:

Reviewer #1 (Remarks to the Author):

I am satisfied with revised version of the manuscript and recommend for publication.

Reviewer #2 (Remarks to the Author):

The authors adequately addressed my comments in the revised manuscript, and I agree that some of the recommended experiments is more appropriate for future work. Given the extensive data already included in the manuscript, I think the authors have convincingly supported their major claim and therefore recommend the publication of this work.

Reviewer #3 (Remarks to the Author):

I think the authors have successfully addressed all three reviewers concerns. To my opinion the manuscript should be accept for publication into Nature Communications

Elena Casacuberta, PhD

CSIC Scientist

Institut of Evolutionary Biology, IBE

CSIC-Pompeu Fabra University (UPF)

Passeig Marítim de la Barcelona, 37-49.

E-08003 Barcelona (Spain)

Tel:+34 932309637